# Improving Defense Mechanisms for Subgraph-Structure Membership Inference Attacks

## Abstract

Graph neural networks (GNNs) are of significant importance in diverse real-world applications since they leverage powerful graph learning techniques to solve problems pertaining to social network mining and medical data analysis. Despite their practical relevance, GNNs remain vulnerable to adversarial attacks such as membership inference attacks (MIAs) which pose privacy risks by revealing whether specific data records were part of the training set of the model. While most existing research has focused on designing defense mechanisms for known node-level MIAs, and in particular, for determining if a certain node was used during training, only limited attention has been paid to subgraph-structure MIA (SMIA) problems. SMIA methods seek to infer whether a set of nodes forms a particular target structure of interest (such as a graph motif, e.g., clique or multi-hop path) in the training graph. The main contributions of our work are three-fold. The first is a novel robust defense mechanism for GNNs against SMIA attacks. It combines an alternating train-test schedule with a flattening strategy to mitigate the attacks. The second contribution is a new end-to-end SMIA attack model that outperforms existing attacks by using multiset functions to generate learnable embeddings for collections of nodes. Extensive simulations reveal that the new attack model outperforms prior state-of-the-art attack models on GNNs by $12.31\%$ across four datasets when no defense mechanism is present. With the new defense mechanism, one can achieve an average decrease of $14.30\%$ in the attack AUROC and an $10.05\%$ improvement in target model utility compared to classical defenses, even when using the improved attack scheme. The third contribution is a study that shows that our defense mechanism extends to node-level MIAs as well, offering similar improvements in attack resistance and utility.

## 1 Introduction

Graph neural networks (GNNs) have emerged as indispensable learning modalities for diverse real-world problems, ranging from social network mining and recommendation system design to biological data analysis (Wu et al., 2021b; Zhang et al., 2024). For example, GNNs have been used to improve personalized search and recommendations for customers on e-commerce platforms (e.g., AliGraph at Alibaba (Zhu et al., 2019) and GIANT at Amazon (Chien et al., 2021a)) and to perform inference and prediction on social networks (e.g., PinnerSage at Pinterest (Pal et al., 2020) and LiGNN at LinkedIn (Borisyuk et al., 2024)). GNNs, unlike standard neural networks, make full use of both the discrete graph topology and node and edge features via special embedding methods based on graph convolutions or random walks (Wu et al., 2020). For example, graph convolutions allow GNNs to generate informative embeddings that are well-suited for different downstream tasks.

Despite the successful deployment of GNNs, several weaknesses of GNN models have been pointed out in the literature. One major concern pertains to data privacy, which is becoming an issue of ever increasing importance. GNNs have exhibited privacy vulnerabilities to various attacks (Sun et al., 2023) such as membership inference attacks (MIAs) (Shokri et al., 2017; Hu et al., 2022; Olatunji et al., 2021), whose aim is to determine if a certain sample is used in model training; attribute inference attacks (AIAs), whose focus is on inferring statistical information about the data, such as the number of nodes and edges (Gong & Liu, 2018) and others. Among all existing attacks, the most commonly observed and studied attacks are MIAs. As already pointed out, MIAs have the goal to reveal whether a given record is part of the training dataset used to build a specific target model,

and are usually based on the model itself, the record and information about the dataset. Typically, MIAs use prediction logits of shadow models to train attack models, where the shadow training data is obtained either through inference of the target model or through access to a potentially noisy version of the original training dataset. Clearly, successful attacks lead to unacceptable information leakage through the model. For example, if a GNN is trained on nodes belonging to a private group within a large social network – e.g., a support group for sensitive medical issues, successful MIAs on GNNs can reveal both the patient identity and their medical condition. As MIAs exploit the differences of the outputs of target model on training and test datasets, most defense mechanisms work towards suppressing the common patterns that quality attacks rely on (Shokri et al., 2017; Jia et al., 2019; Choquette-Choo et al., 2021; Hayes et al., 2017; Leino & Fredrikson, 2020; Salem et al., 2018; Kaya & Dumitras, 2021; Yu et al., 2021; Wang et al., 2020; Chen et al., 2022).

Recent studies have focused on node (or edge) MIAs for node (or edge) classification downstream tasks (Olatunji et al., 2021; Wu et al., 2021a; He et al., 2021; Conti et al., 2022). Compared to these standard MIAs on GNNs, little attention has been paid to subgraph-structure MIAs (SMIAs) (Wang & Wang, 2024). SMIAs can lead to an especially problematic form of privacy leakage, where attackers can infer not only if certain nodes were present in the training graph but also if there were certain relationships between them. These relationships are usually captured via graph motives (triangles, cliques, paths etc). For instance, in a medical data network, cliques may involve members of the same family and indicative of genetic/familial diseases. In this case, SMIA attacks can compromise not only individual medical histories, but – by association – whole family medical conditions. In comparison, standard MIA methods can only reveal if the nodes were used for training a node classifier, but not what the relationships between themselves and other nodes in the graph are (one may think it plausible to perform individual node MIA attacks and then use link inference attacks to infer the subgraph induced by these nodes, but this process is both ineffective and usually of poor utility). It is also important to point out that SMIA attacks differ from subgraph inference attacks (SIAs) (Zhang et al., 2022) whose goal is to determine if a certain type of subgraph is present in the training dataset (without revealing the nodes that constitute the subgraph).

The only prior line of work on SMIAs is Wang & Wang (2024). There, the authors propose an attack method based on the use of similarities between posterior vectors of the target nodes which allows them to generate training data for the attack model. On the defense side, the authors calculate the importance of each dimension of the node embeddings using the SHAP algorithm (Scott et al., 2017), and add noise to the least important dimensions in an attempt to balance out the quality of the defence against SMIAs and the utility of the model. However, this attack and defense mechanisms have several notable limitations. First, the similarity metrics used for the attack are fixed and cannot be adapted even when the data distribution has changed. Second, the attack performs well only on homophilic graphs, but as we subsequently show, offers poor performance on heterophilic graphs. Third, the similarity calculations split the attack process into two different parts, preventing an end-to-end approach, which results in an increase of the complexity of practical implementation. Finally, the defense still relies on the addition of noise to the embeddings, which inevitably compromises the model utility. To address these issues, we propose both a new attack and improved defense system that can counter both the attack of the original SMIA and our improved attack. The gist of our approach is to replace the fixed similarity calculation with learnable multiset functions that allow for an end-to-end attack that dynamically adapts the training data generation process to the dataset, thereby offering excellent performance on both homophilic and heterophilic datasets. Furthermore, we propose a novel two-stage defense (TSD) strategy followed by flattening (Chen et al., 2022) that does not significantly sacrifice model utility but provides stronger defense capabilities compared to noise addition. The key intuition behind the defense is to obfuscate the posterior distributions via *controlled overfitting* and to add flattening noise that can also obfuscate the graph information (note that flattening noise cannot be directly "translated" to adding noise to node embeddings). Under the TSD approach, the attack approach learned on GNNs without defense mechanisms cannot be easily adapted to those which use defense, leading to a drop in the attack performance. In parallel, flattening (Chen et al., 2022) allows us to increase the variance of the training loss distribution and provides another mechanism for ensuring different posterior distribution for test and trainsets (since different loss distributions lead to different posterior distributions). A detailed description of the approach is delegated to Section 4.

To demonstrate the utility of our novel SMIA attack and defense, we adopt the following standard modeling assumptions. First, we focus on black-box attacks. Second, unlike some other approaches

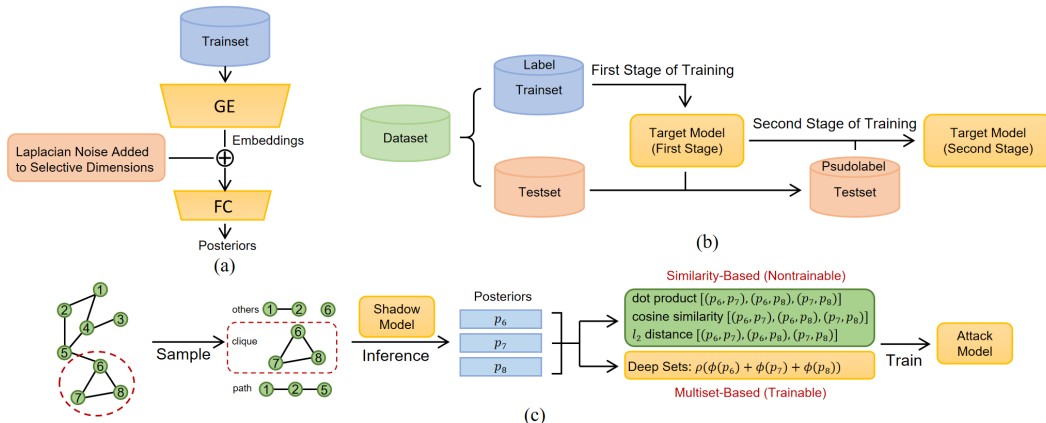

Figure 1: (a) Diagram of the defense method SHAP-based noise addition (SHNA) proposed by Wang & Wang (2024). Here GNNs are split into two parts: graph encoder (GE) that embeds node features and graph topology as node embeddings, and fully connected (FC) layer that transform node embeddings into posteriors. (b) Diagram of the two-stage training schedule that leads to an improved defense against SMIAs. The key idea behind the TSD approach is to use the predicted labels of test nodes from the first stage as psudolabels and switch the train and testset in the second stage. During both training stages, flattening is performed as well to alleviate overfitting (see Section 4). (c) Comparison of standard SMIA (similarity-based) and our new end-to-end SMIA approach with improved and generalized attacks. The standard SMIA model uses pairwise similarity of posteriors to generate data for training the attack model. In contrast our end-to-end SMIA replaces similarity metrics with multiset functions that directly deal with posteriors in a permutation invariant fashion. This allows the attacker to be trained end-to-end from data preparation to model training.

that only describe attacks for specific GNN architectures or fixed defense mechanisms (Olatunji et al., 2021), we design our defense strategy to be generally applicable to different graph learning paradigms, datasets and resistant to both standard and improved SMIA attacks. Our extensive experiments, performed on three homophilic (CiteSeer, Facebook, LastFM) and one heterophilic datset (Chameleon), coupled with GCN, GAT, SGC and GPRGNN networks (Kipf & Welling, 2016; Velickovic et al., 2017; Wu et al., 2019; Chien et al., 2020), reveal that our end-to-end attack model outperforms the standard SMIA by $12.31\%$ when no defense is used. We also demonstrate an average $14.30\%$ decrease in attack AUROC and an $10.05\%$ improvement in target model utility trained with our new defense technique compared to classical defenses, and under the improved attack.

## 2 RELATED WORKS

Due to space limitations, we only provide a brief summary of related results and delegate a more detailed discussion to Appendix A.

**MIAs.** The concept of MIA was first proposed in Homer et al. (2008) and later extended in various directions, ranging from white-box settings (Nasr et al., 2019; Rezaei & Liu, 2021; Melis et al., 2019; Leino & Fredrikson, 2020) to black-box setting (Shokri et al., 2017; Salem et al., 2018; Song & Mittal, 2020; Li & Zhang, 2021; Choquette-Choo et al., 2021; Carlini et al., 2022). Upon identification of the informative features (e.g., posterior predictions, loss values, gradient norms, etc.) that reveal the sample membership, the attacker can choose to learn either a binary classifier (Shokri et al., 2017) or metric-based decisions (Yeom et al., 2018; Salem et al., 2018) from a shadow model trained on a shadow dataset to extract patterns within features of the training samples to determine the membership. The description of a standard MIA pipeline is available in Appendix C.

**Defense Against MIAs.** As MIAs exploit the behavioral differences of the target model on trainsets and testsets, most defense mechanisms work by suppressing the common patterns among the two. Frequently used defense methods include confidence score masking (Shokri et al., 2017; Jia et al., 2019; Yang et al., 2020; Li et al., 2021; Choquette-Choo et al., 2021; Hanzlik et al., 2021), regularization (Hayes et al., 2017; Salem et al., 2018; Leino & Fredrikson, 2020; Wang et al., 2020;

Choquette-Choo et al., 2021; Kaya & Dumitras, 2021; Yu et al., 2021; Chen et al., 2022), knowledge distillation (Shejwalkar & Houmansadr, 2020; Tang et al., 2022), and differential privacy (Naseri et al., 2020; Saeidian et al., 2021). For example, confidence score masking aims to hide the true prediction vector returned by the target model and it thus mitigates the effectiveness of MIAs by providing only top-$k$ logits per inference (Shokri et al., 2017), or adding noise to the prediction vector in an adversarial manner (Jia et al., 2019). Regularization aims to reduce the degree of over-fitting of target models to mitigate MIAs (Choquette-Choo et al., 2021; Hayes et al., 2017; Leino & Fredrikson, 2020; Salem et al., 2018; Kaya & Dumitras, 2021; Yu et al., 2021; Wang et al., 2020; Chen et al., 2022). Knowledge distillation aims to transfer the knowledge from an unprotected model to a protected model (Shejwalkar & Houmansadr, 2020), while differential privacy (Saeidian et al., 2021) protects membership information via noise injection and offers theoretical performance guarantees, at the cost of substantial utility drop.

**MIAs and Defense Strategies for GNNs.** There are a handful of research that focuses on extending MIA and corresponding defense mechanisms to graph learning framework. Olatunji et al. (2021) analyzed graph MIA in two settings (train on subgraph, test on subgraph/full), and proposed the LBP defense based on the confidence score masking idea, He et al. (2021) proposed zero-hop and two-hop attacks designed for inductive GNNs, Wang & Wang (2023) studied the link membership inference problem in an unsupervised fashion, and Chen et al. (2024) developed MaskArmor based on masking and distillation technique. Besides node and edge MIAs, Zhang et al. (2022); Wang & Wang (2024) also explored the subgraph attacks along with perturbation-based defense mechanism.

$k$**-node Structure Membership Inference ($k$-SMIA)** is a new form of a privacy attack (Wang & Wang, 2024). The aim of this attack is to determine whether a collection of $k$ nodes within the training set belongs to a subgraph structure of interest (e.g., path or clique). $k$-SMIA hence uses a new definition of membership: members are substructures of $k$ nodes that form a relevant topology. The original SMIA work outlines a novel black-box SMIA attack that combines training a three-label classifier with training shadows. This attack outperforms node-level MIAs followed by link prediction. The defense against SMIA relies on perturbing embedding of the nodes with the smallest "contribution" to the accuracy of the model and it results in a performance comparable to that offered by differential privacy. Despite these positive initial findings, one has to point out that a performance matched by differential privacy methods may not be desirable, since the latter is usually significantly reduced compared to defenseless models. Note that $k$-SMIA still constitutes an attack against node-level models, so that the downstream task of the target model remains node classification.

## 3 PROBLEM FORMULATION

In this paper we focus on the downstream task as supervised node classifications; nevertheless, our method is applicable to different graph learning scenarios. Let $\mathcal{G} = (X, A, Y, \mathcal{V}^{\text{Train}}, \mathcal{V}^{\text{Test}})$ denote the graph dataset with node features $X \in \mathbb{R}^{n \times d}$, adjacent matrix $A \in \mathbb{R}^{n \times n}$, one-hot encoded node labels $Y \in \mathbb{R}^{n \times C}$, trainset $\mathcal{V}^{\text{Train}}$ and testset $\mathcal{V}^{\text{Train}}$. Here $n$ is the number of nodes, $d$ is the feature dimension, $C$ is the number of classes, $\mathcal{V}^{\text{Train}}$ and $\mathcal{V}^{\text{Test}}$ are disjoint and $|\mathcal{V}^{\text{Train}}| + |\mathcal{V}^{\text{Test}}| = n$. We later on use $Y^{\text{Train}}$ to denote the labels of $\mathcal{V}^{\text{Train}}$, and $\hat{Y}^{\text{Test}}$ to denote the predicted labels of $\mathcal{V}^{\text{Test}}$. Since $X$ and $A$ are already known during training, the goal of $k$-**SMIA** is determing the **subgraph-substructure membership**: given a set of $k$ nodes $\mathcal{V}_{att} \subset \mathcal{V}^{\text{Train}} \cup \mathcal{V}^{\text{Test}}$, determine if $\mathcal{V}_{att}$ forms either a $k$-clique or a $(k-1)$-hop path in $\mathcal{G}$ (member) or not (non-member). Meanwhile, the goal of **MIA** is determing the **label membership**: given a node $v \in \mathcal{V}^{\text{Train}} \cup \mathcal{V}^{\text{Test}}$, determine if $v \in \mathcal{V}^{\text{Train}}$ (member) or not (non-member). Following the practice of SMIA and MIA, we also need a shadow dataset $\mathcal{G}_s = (X_s, A_s, Y_s, \mathcal{V}_s^{\text{Train}}, \mathcal{V}_s^{\text{Test}})$ to train the shadow model, and $\mathcal{G}_s$ can be different from $\mathcal{G}$.

## 4 THE TSD METHOD

To defend against SMIAs, we introduce a *two-stage defense* method (TSD) to train *target* GNNs, depicted in Algorithm 1. The key intuition behind our method is to change the posterior distribution.

The first stage of TSD involves using $\mathcal{G}$ to train a model checkpoint $M_1(\theta_1)$ with parameters $\theta_1$. We adopt a flattening strategy in the first stage as a form of regularization, inspired by (Chen et al., 2022). The flattening is implemented by transforming hard labels (one-hot) to soft labels (probability vectors) when the loss on the trainset falls below a threshold $\alpha$. For simplicity, we assign the value

$\beta$ to the ground-truth class, and $\frac{1-\beta}{C-1}$ to the other classes. Here $\alpha, \beta$ are hyperparameters while $C$ denotes the number of classes. Note that we only use soft labels to compute the loss when the loss is small enough to keep the model utility as high as possible. The key of flattening is to increase the mean and variance of the training loss distribution while introducing noise to the label distribution.

---

**Algorithm 1** TSD Training Procedure

---

**Input:** Dataset $\mathcal{G} = (X, A, Y, \mathcal{V}^{\text{Train}}, \mathcal{V}^{\text{Test}})$, number of training epochs $E$, learning rate $\gamma$, number of classes $C$, loss threshold $\alpha$, flattening parameter $\beta$.
**Output:** Second stage target model $M_2(\theta_2)$
**First stage:**
Perform random initialization for the first stage model $M_1(\theta_1)$
**for** epoch $\in [1, E]$ **do**
    **if** loss $L(M_1(\theta_1), Y^{\text{Train}}) \geq \alpha$ **then**
        $\theta_1 \leftarrow \theta_1 - \gamma \cdot \nabla_{\theta_1} L(M_1(\theta_1), Y^{\text{Train}})$
    **else**
        Construct soft labels $S^{\text{Train}}$ where $s_c^{\text{Train}} =$
        $\begin{cases} \beta, & \text{if } y_c^{\text{Train}} = 1; \\ (1-\beta)/(C-1), & \text{otherwise} \end{cases}$
        $\theta_1 \leftarrow \theta_1 - \gamma \cdot \nabla_{\theta_1} L(M_1(\theta_1), S^{\text{Train}})$
    **end if**
**end for**
**Second stage:**
Initialize the second stage model $M_2(\theta_2)$ with $M_1(\theta_1)$, and generate pseudolabels $\hat{Y}^{\text{Test}}$ for the original testset by inference via $M_1(\theta_1)$
**for** epoch $\in [1, E]$ **do**
    **if** loss $L(M_2(\theta_2), \hat{Y}^{\text{Test}}) \geq \alpha$ **then**
        $\theta_2 \leftarrow \theta_2 - \gamma \cdot \nabla_{\theta_2} L(M_2(\theta_2), \hat{Y}^{\text{Test}})$
    **else**
        Construct soft labels $S^{\text{Test}}$ where $s_c^{\text{Test}} =$
        $\begin{cases} \beta, & \text{if } \hat{y}_c^{\text{Test}} = 1; \\ (1-\beta)/(C-1), & \text{otherwise} \end{cases}$
        $\theta_2 \leftarrow \theta_2 - \gamma \cdot \nabla_{\theta_2} L(M_2(\theta_2), S^{\text{Test}})$
    **end if**
**end for**

---

The second stage of TSD is similar to the first one, with the main difference that we instead use $(X, A, \hat{Y}^{\text{Test}})$ for training. The pseudolabels $\hat{Y}^{\text{Test}}$ are generated via inference of the checkpoint $M_1(\theta_1)$ on the testset. Instead of random initialization, we use checkpoint initialization for the second stage model $M_2(\theta_2)$ to resume training. The subsequent training process also proceeds with flattening, and $M_2(\theta_2)$ is the final output target model. The role of the second stage is to include the testset into training, even without having access to their groundtruth labels $Y^{\text{Test}}$. In this case, the testset is also "trained", as it undergoes through the same process as the trainset.

**SHAP-based noise addition (SHNA) defense.** The SHNA defense (Wang & Wang, 2024) aims to weaken the attack by introducing noise into the node embeddings. To reduce the impact of noise on the model performance, SHNA uses the SHAP algorithm (Scott et al., 2017) to quantify the contribution of each embedding dimension to the final classification accuracy, and selectively adds noise to the $l = \lfloor r \times d \rfloor$ dimensions with the smallest contributions, where $r \in (0, 1]$ is a hyperparameter, and $d$ is the embedding dimension. The added noise is Laplacian, with scale $b$ and mean $\mu$.

**Comparison with SHNA.** The reasons why TSD can outperform SHNA are two-fold. First, as we explain in Section 5, the attacker targets the posterior distributions directly rather than the node embeddings. Therefore, adding noise to the node embeddings is only an *indirect* method of perturbing the attack model's input, and the relationship between the noise scale and the attackers performance is unclear. In contrast, TSD directly alters the posterior distribution, as evidenced by the significant change in the loss distribution. Figure 2 illustrates this point: without TSD (standard training), the average training loss is lower than the testing loss. With TSD, however, the discrepancy in the loss distribution is notably reduced. Additionally, we conduct another simulation to examine the correlation between attack performance and loss distribution. Note that when a clique exists in the graph, three scenarios are possible: (1) all nodes belong to the trainset; (2) nodes belong to both the train and testset; and (3) all nodes belong to the testset. The attack accuracy for triangle identification in these cases is $0.9722$, $0.9347$, and $0.8526$, respectively. These results confirm that the attack accuracy is positively correlated with how well the posterior distribution is learned.

# 5 END-TO-END SMIA: A MULTISET FUNCTION APPROACH

To demonstrate that our defense mechanism is effective against various types of SMIAs, we evaluate it using both the similarity-based attacks introduced by Wang & Wang (2024) and a novel attack based on multiset functions. We begin by reviewing the similarity-based attacks, highlighting their strengths and limitations, before introducing our new end-to-end attack scheme.

**Similarity-Based Attacks.** The core of similarity-based SMIA lies in processing the shadow model's output to generate data for attack model training. For example, for a $k = 3$ clique, the shadow model provides three posterior vectors for three nodes. Pairwise similarities between these vectors are computed and sorted in ascending order to form a new vector. This process is repeated for three different similarity metrics, namely the dot product, cosine similarity, and $\ell_2$ distance. The sorted vectors for each metric are concatenated to create the training data for the attack model along with the structure labels (see also Appendix B).

While similarity-based attacks have been shown to outperform other link-level attacks for SMIA, they have limitations. First, the choice of similarity metrics is heuristic and may not be effective in all scenarios. Second, the computational complexity is $O(k^3)$[1], which is prohibitive for large $k$.

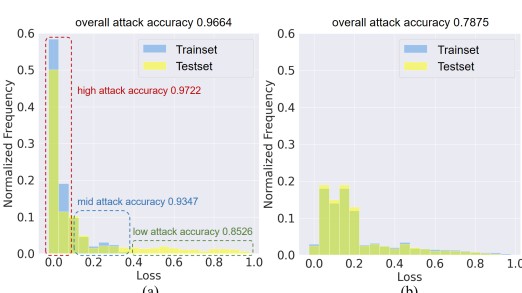

**Multiset Function-Based Attacks.** To address the limitations of similarity-based attacks, we propose using multiset functions to aggregate posterior vectors, rather than relying on heuristic similarity measures. The primary advantage of multiset functions is that they can directly process posterior vectors without additional preprocessing. Furthermore, multiset

Figure 2: Comparison of training and test loss distribution under (a) standard training; (b) TSD training. The simulation is conducted on Chameleon with NLGCN as the backbone. The attack is our end-to-end SMIA.

functions treat inputs as sets, ensuring that the output remains unaffected by the input order. We implement the multiset function using Deep Sets (Zaheer et al., 2017), which has proven effective in other contexts requiring permutation invariance, such as hypergraph GNNs (Chien et al., 2021b). For example, consider three nodes $v_1, v_2, v_3$ with corresponding posterior distributions $p_1, p_2, p_3$. The final embedding used for training the attacker is computed as $\rho(\phi(p_1) + \phi(p_2) + \phi(p_3))$, where $\rho$ and $\phi$ are neural networks (specifically, two-layer MLPs in our simulations), which makes the attack model trainable in an *end-to-end* fashion. Based on the performance guarantees of Deep Sets and the universal approximation theorem, it is straightforward to show that the similarity-based approach is a special case of the multiset function-based method.

## 6 EXPERIMENTS

### 6.1 EXPERIMENTS PERTAINING TO SMIA DEFENSE

**Datasets and GNN Baselines.** We train four GNN (GCN (Kipf & Welling, 2016), GAT (Velickovic et al., 2017), SGC (Wu et al., 2019), GPRGNN (Chien et al., 2020)) on three homophilic datasets (CiteSeer, Facebook (Yang et al., 2023), LastFM (Rozemberczki & Sarkar, 2020)), and four GNN (NLGCN, NLGAT, NLMLP (Wang et al., 2018), GPRGNN (Chien et al., 2020)) on one heterophilic datasets (Chameleon (Rozemberczki et al., 2021)). Detailed experimental setups, properties and statistics of the datasets are available in Appendix F.

**Evaluation Metrics.** We follow previous literature to use the following two metrics for evaluation. We report classification accuracy of the target model on the testset (CA) to measure model utility, and AUROC scores of the attack model (AU), which is a widely used approach in the field of MIA (Carlini et al., 2022). Note that good defenses should lead to large CAs and small AUs.

#### 6.1.1 COMPARISON OF SMIA DEFENSE METHODS AND ATTACK METHOD

Table 1 and 2 present a comparison of End2end-SMIA with Standard-SMIA, as well as the defense performance of TSD and SHAN. Additional results can be found in Appendix G. Standard-SMIA refers to similarity-based attacks while End2end-SMIA refers to the multiset function-based attack. For simplicity, we refer to them as S-SMIA and E-SMIA, respectively. In the experiments, SHAN uses the noise addition strategy from the original work for homophilic datasets, $r = 0.4$ and $b = 0.3$, and for the heterophilic dataset, $r = 0.1$ and $b = 0.01$.

---

[1] $O(k^2)$ pairs with $O(k)$ computational complexity per pair.

Table 1: 3-SMIA attack performance comparison between Standard-SMIA and End2end-SMIA & Defense performance comparison between SHNA and TSD. Compared to SHNA, TSD achieves a decrease in attack AUROC by $16.64\%$ against Standard-SMIA & $14.30\%$ against End2end-SMIA, and increases in utility performance by $10.05\%$ (on average). Full table with variance information is available in Appendix Table 8.

| Dataset | Models | CA(SHNA) | CA(TSD) | S-SMIA AU (SHNA) | S-SMIA AU (TSD) | E-SMIA AU(SHNA) | E-SMIA AU(TSD) |
|---|---|---|---|---|---|---|---|
| CiteSeer | GCN | 0.6887 | **0.7729** | 0.9157 | **0.7767** | 0.9684 | **0.8703** |
| | GAT | 0.7082 | **0.7548** | 0.8784 | **0.7145** | 0.9336 | **0.8519** |
| | SGC | 0.6982 | **0.7454** | 0.9015 | **0.7978** | 0.9862 | **0.8904** |
| | GPRGNN | 0.6736 | **0.7864** | 0.8222 | **0.5969** | 0.9006 | **0.7364** |
| Facebook | GCN | 0.6028 | **0.7095** | 0.7143 | **0.5463** | 0.7942 | **0.6127** |
| | GAT | 0.5321 | **0.6435** | 0.6854 | **0.5382** | 0.8229 | **0.6068** |
| | SGC | 0.5011 | **0.6423** | 0.7028 | **0.5253** | 0.7903 | **0.5945** |
| | GPRGNN | 0.5529 | **0.6544** | 0.6512 | **0.5562** | 0.8081 | **0.6213** |
| LastFM | GCN | 0.8256 | **0.8523** | 0.9215 | **0.8337** | 0.9825 | **0.8702** |
| | GAT | 0.8379 | **0.8662** | 0.8528 | **0.8664** | 0.9274 | **0.8821** |
| | SGC | 0.8154 | **0.8400** | 0.9305 | **0.8835** | 0.9840 | **0.9060** |
| | GPRGNN | 0.8320 | **0.8577** | 0.9004 | **0.8268** | 0.9543 | **0.8687** |
| Chameleon | NLGCN | 0.6373 | **0.6725** | 0.8904 | **0.4538** | 0.9196 | **0.7875** |
| | NLGAT | 0.6437 | **0.6835** | 0.7025 | **0.5611** | 0.8848 | **0.7958** |
| | NLMLP | 0.5010 | **0.5484** | 0.7128 | **0.5342** | 0.8152 | **0.7057** |
| | GPRGNN | 0.6637 | **0.6835** | 0.7191 | **0.6259** | 0.8929 | **0.7711** |

Table 2: 4-SMIA attack performance comparison between Standard-SMIA and End2end-SMIA & Defense performance comparison between SHNA and TSD. Compared to SHNA, TSD achieves a decrease in attack AUROC by $16.56\%$ against Standard-SMIA & $13.89\%$ against End2end-SMIA, and increases in utility performance by $10.05\%$ (on average). Full table with variance information is available in Appendix Table 9.

| Dataset | Models | CA(SHNA) | CA(TSD) | S-SMIA AU(SHNA) | S-SMIA AU(TSD) | E-SMIA AU(SHNA) | E-SMIA AU(TSD) |
|---|---|---|---|---|---|---|---|
| CiteSeer | GCN | 0.6887 | **0.7729** | 0.9685 | **0.9013** | 0.9786 | **0.9267** |
| | GAT | 0.7082 | **0.7548** | 0.9718 | **0.8868** | 0.9891 | **0.9191** |
| | SGC | 0.6982 | **0.7454** | 0.9839 | **0.9316** | 0.9955 | **0.9442** |
| | GPRGNN | 0.6736 | **0.7864** | 0.9010 | **0.8523** | 0.9692 | **0.8894** |
| Facebook | GCN | 0.6028 | **0.7095** | 0.7377 | **0.5308** | 0.7964 | **0.6033** |
| | GAT | 0.5321 | **0.6435** | 0.6732 | **0.5027** | 0.7436 | **0.5978** |
| | SGC | 0.5011 | **0.6423** | 0.6992 | **0.5097** | 0.7520 | **0.6325** |
| | GPRGNN | 0.5529 | **0.6544** | 0.6520 | **0.4978** | 0.7090 | **0.5885** |
| LastFM | GCN | 0.8256 | **0.8523** | 0.9649 | **0.8296** | 0.9972 | **0.8561** |
| | GAT | 0.8379 | **0.8662** | 0.8936 | **0.7442** | 0.9354 | **0.7654** |
| | SGC | 0.8154 | **0.8401** | 0.9574 | **0.8184** | 0.9979 | **0.8270** |
| | GPRGNN | 0.8320 | **0.8577** | 0.9144 | **0.8044** | 0.9844 | **0.8444** |
| Chameleon | NLGCN | 0.6373 | **0.6725** | 0.7663 | **0.5668** | 0.9952 | **0.8346** |
| | NLGAT | 0.6437 | **0.6835** | 0.7849 | **0.5901** | 0.9816 | **0.8409** |
| | NLMLP | 0.5010 | **0.5484** | 0.7531 | **0.6076** | 0.9186 | **0.8359** |
| | GPRGNN | 0.6637 | **0.6835** | 0.8358 | **0.5924** | 0.9486 | **0.8127** |

When comparing the attack AUROC on the same dataset, using the same model and defense method, End2end-SMIA is noticeably better than Standard-SMIA, especially on the heterophilic Chameleon. This is due to the trainable Deep Sets, which extract structural information of the node sets from their posterior vectors, significantly reducing the learning difficulty for the attack model. Additionally, whereas the similarity metrics chosen in Standard-SMIA may not be well-suited to the attacked node sets, Deep Sets can continuously adjust during training to offer improved performance.

When comparing TSD and SHAN under the same conditions, we observe that TSD offers superior defensive capabilities while maintaining higher classification accuracy for the target model. This is because TSD alters the posterior distribution through an additional training phase and the introduction of flattening. These two methods are applied only after the target model has already completed learning from the data, so the classification performance of the target model remains largely unaffected. In contrast, SHAN aims to minimize the impact on the target model's classification performance by adding noise only to the least important dimensions of the embeddings. However, this also means that the noise's ability to disrupt the attack decreases due to the reduced importance of those dimensions. Therefore, it is difficult to achieve both high model utility and strong defense ability, regardless of how the noise is added.

### 6.1.2 DATA TRANSFER

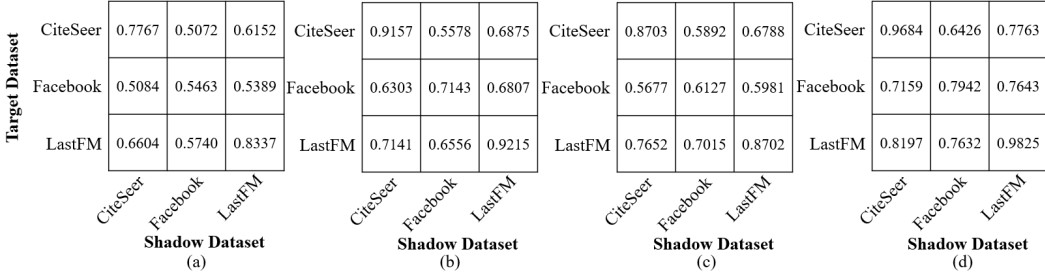

Figure 3: Result for 3-SMIA Attack AUROC on GCN under data transfer setting for (a) TSD against Standard-SMIA; (b) SHAN against Standard-SMIA; (c) TSD against End2end-SMIA; and (d) SHAN against End2end-SMIA.

In practical applications, the training data of the target model is typically unknown to the attacker. Therefore, to train a shadow model, attackers often use publicly available datasets as training data, which usually have different distributions from the training data of the target model. To better evaluate the effectiveness of the TSD defense method, we set up an experimental scenario that resembles real-world SMIA, where experiments are conducted for the data transfer setting. *Data transfer* refers to the situation where the training data for the target model and the shadow model come from different datasets. Evidently, when the training data for the target and shadow models come from the same dataset, the attack on the target model is the strongest, corresponding to the lower bound of the defense method's effectiveness.

Figure 3 presents the results obtained using GCN as both the target and shadow model under the 3 SMIA attack. The experimental settings remain the same as described in Section 6.1.1. The results demonstrate the continued effectiveness of the TSD defense method in the data transfer setting. Across all dataset combinations, TSD consistently outperformed SHAN, indicating that TSD can maintain excellent performance against SMIA attacks that may occur in real-world scenarios. Additionally, it is worth noting that End2end-SMIA exhibits stronger attack performance compared to Standard-SMIA, which can be attributed to the powerful generalization capability of DeepSets.

### 6.1.3 ABLATION STUDY OF TSDS

Table 3: Ablation study for the contributors to the utility/attack gains/mitigation for TSD under 3-SMIA Attack.

| Method | Dataset | GNN Models | Classify Acc | Attack AUROC |
|---|---|---|---|---|
| Standard Training | CiteSeer | GCN | $0.8012 \pm 0.0093$ | $0.9745 \pm 0.0084$ |
| | Facebook | GCN | $0.7353 \pm 0.0085$ | $0.8759 \pm 0.0091$ |
| | LastFM | GCN | $0.8823 \pm 0.0088$ | $0.9841 \pm 0.0094$ |
| | Chameleon | NLGCN | $0.7052 \pm 0.0085$ | $0.9664 \pm 0.0084$ |
| Flattening (One-Stage) | CiteSeer | GCN | $0.7956 \pm 0.0082$ | $0.9428 \pm 0.0081$ |
| | Facebook | GCN | $0.7291 \pm 0.0080$ | $0.8249 \pm 0.0084$ |
| | LastFM | GCN | $0.8762 \pm 0.0096$ | $0.9520 \pm 0.0075$ |
| | Chameleon | NLGCN | $0.7009 \pm 0.0094$ | $0.9187 \pm 0.0093$ |
| Two-Stage (without Flattening) | CiteSeer | GCN | $0.7804 \pm 0.0087$ | $0.9013 \pm 0.0081$ |
| | Facebook | GCN | $0.7168 \pm 0.0080$ | $0.6644 \pm 0.0091$ |
| | LastFM | GCN | $0.8610 \pm 0.0076$ | $0.9008 \pm 0.0091$ |
| | Chameleon | NLGCN | $0.6863 \pm 0.0085$ | $0.8324 \pm 0.0074$ |
| TSD (Two-Stage & Flattening) | CiteSeer | GCN | $0.7729 \pm 0.0021$ | $0.8703 \pm 0.0075$ |
| | Facebook | GCN | $0.7095 \pm 0.0034$ | $0.6127 \pm 0.0096$ |
| | LastFM | GCN | $0.8523 \pm 0.0027$ | $0.8702 \pm 0.0088$ |
| | Chameleon | NLGCN | $0.6725 \pm 0.0054$ | $0.7875 \pm 0.0098$ |

Our TSD method differs from the conventional training methods in two aspects: (1) the use of flattening operations; and (2) two-stage training. To demonstrate their roles in enhancing defense capability, we conducted the following ablation experiments.

In the experiments, we examined four variants: (1) standard training; (2) two-stage (without flattening); (3) flattening (one-stage); and (4) TSD. Here, standard training refers to training a target model only on the trainset; two-stage trains a target model in a train-test alternate fashion, equivalent to TSD without flattening; flattening is the same as described in Section 4, combined with one-stage training. Clearly, TSD is two-stage combined with flattening. We conduct our experiments using End2end-SMIA.

Table 3 presents the results of the ablation study for these four variants on four datasets and one GNN backbone – GCN. Additional results can be found in Appendix I. The findings indicate that the primary source of improvement for TSD is the two-stage training technique. Moreover, compared to the defense improvement brought by flattening, its impact on model utility is acceptable.

## 6.2 EXPERIMENTS PERTAINING TO MIA DEFENSE

TSD is also capable of defending against node-level MIA. Previous studies on MIA (Homer et al., 2008; Shokri et al., 2017; Salem et al., 2018; Song & Mittal, 2020) have pointed out that the key to a successful MIA attack is to allow overfitting in the target model. Since TSD has the ability to reduce overfitting, it should intuitively have the capability to defend against MIA: the two-stage training process in TSD can help reduce the gap between the average losses of training and testing nodes, while the use of flattening reduces the difference in the variance of loss distributions between the training and testing nodes. We demonstrate the effectiveness of TSD's defense by comparing it with other defense methods. We choose two representative defense methods, Laplacian Binned Posterior Perturbation (LBP) (Olatunji et al., 2021) on GNNs and Distillation for Membership Privacy (DMP) (Shejwalkar & Houmansadr, 2020) on graphless models, as our defense baselines. LBP is the state-of-the-art defense method for GNNs and it works by adding Laplacian noise to the posterior before it is released to the user. To reduce the amount of noise needed to distort the posteriors, LBP does not add noise to each element of the posterior, but to binned posterior. In our experiments, we first randomly shuffle the posteriors and then assign each posterior to a partition/bin. The total number of bins $N$ is predefined based on the number of classes. For each bin, we sample noise with scale $b$ from the Laplace distribution. The sampled noise is added to each element in the bin. After the noise added to each bin, we restore the initial positions of the noisy posteriors and release them. Appendix Table 13 shows the best set of parameters for LBP that we used in our experiments.

On the other hand, we adapted DMP to the case of GNN training. DMP consists of three phases, namely pre-distillation, distillation and post-distillation. The pre-distillation phase trains an unprotected model on a private training data without any privacy protection. Next, in the distillation phase, DMP selects reference data and transfers the knowledge of the unprotected model into predictions om the reference data. Notice that private training data and the reference data have no intersection. Finally, in the post-distillation phase, DMP uses the predictions to train a protected model. Our experiments use the same model structure for the unprotected and protected models. To follow the DMP procedure, we need to further split the trainset into a private dataset and reference dataset, where the private dataset trains the unprotected models, and the reference dataset trains the protected target model. Compared to DMP, TSD can directly train the target model with the full trainset.

In our experiments, the split ratio of trainsets and testsets for TSD and LBP is 1:1, and the split ratio of private datasets, reference datasets and testsets in DMP is 0.45:0.45:0.1. Since MIA is easier to defend against than SMIAs, our experiments are conducted on more complex and diverse datasets. Table 4 and Table 5 shows partial result of our experiments, and the complete results can be found in Appendix J.1 and J.2. The results indicate that our method achieves defense capabilities comparable to LBP and DMP, while achieving better model utility performance. It is important to emphasize that a lower attack AUROC does not necessarily indicate stronger defense. The stronger the defense, the closer the AUROC should be to 0.5, as 0.5 represents random guessing. When the AUROC is less than 0.5, the attackers can flip the prediction results to make the AUROC greater than 0.5. Compared to LBP, TSD significantly improved the model utility. The main reason is that LBP is a perturbation-based method, which can potentially hurt the target model performance significantly. However, TSD achieves defense by alleviating overfitting, which delves deeper into the core issue, instead of adversely affecting target models. In addition, compared to DMP, TSD still achieves

higher model utility. This gain is expected, as TSD not only can make use of the full trainsets, but also utilizes testsets in the second stage, thus enhancing the model's generalization ability.

Table 4: Performance comparison between TSD and LBP. Compared to LBP, TSD has an average increase in utility by 17.28%, with a comparable attack AUROC.

| Dataset | Models | Classify Acc (LBP) | Classify Acc (TSD) | Attack AUROC (LBP) | Attack AUROC (TSD) |
|---------|--------|--------------------|--------------------|--------------------|--------------------|
| PubMed | GCN | $0.6886 \pm 0.0041$ | $\mathbf{0.8381 \pm 0.0023}$ | $\mathbf{0.4998 \pm 0.0050}$ | $0.4990 \pm 0.0048$ |
| | GAT | $0.7631 \pm 0.0037$ | $\mathbf{0.8400 \pm 0.0028}$ | $\mathbf{0.5021 \pm 0.0084}$ | $0.4911 \pm 0.0061$ |
| | SGC | $0.6564 \pm 0.0035$ | $\mathbf{0.8080 \pm 0.0020}$ | $0.5007 \pm 0.0065$ | $\mathbf{0.5005 \pm 0.0057}$ |
| | GPRGNN | $0.7843 \pm 0.0029$ | $\mathbf{0.8553 \pm 0.0014}$ | $\mathbf{0.5003 \pm 0.0038}$ | $0.4967 \pm 0.0034$ |
| Facebook | GCN | $0.5195 \pm 0.0041$ | $\mathbf{0.6778 \pm 0.0049}$ | $0.4912 \pm 0.0021$ | $\mathbf{0.4993 \pm 0.0023}$ |
| | GAT | $0.5460 \pm 0.0037$ | $\mathbf{0.6519 \pm 0.0039}$ | $0.5120 \pm 0.0025$ | $\mathbf{0.5010 \pm 0.0028}$ |
| | SGC | $0.4833 \pm 0.0044$ | $\mathbf{0.6249 \pm 0.0043}$ | $0.4901 \pm 0.0026$ | $\mathbf{0.5004 \pm 0.0031}$ |
| | GPRGNN | $0.4627 \pm 0.0032$ | $\mathbf{0.5890 \pm 0.0035}$ | $0.4807 \pm 0.0031$ | $\mathbf{0.5014 \pm 0.0020}$ |
| Lastfm | GCN | $0.6509 \pm 0.0037$ | $\mathbf{0.8378 \pm 0.0035}$ | $0.5118 \pm 0.0024$ | $\mathbf{0.4971 \pm 0.0022}$ |
| | GAT | $0.7210 \pm 0.0034$ | $\mathbf{0.8683 \pm 0.0032}$ | $0.5136 \pm 0.0031$ | $\mathbf{0.4980 \pm 0.0029}$ |
| | SGC | $0.6395 \pm 0.0040$ | $\mathbf{0.8336 \pm 0.0045}$ | $0.5121 \pm 0.0030$ | $\mathbf{0.4965 \pm 0.0034}$ |
| | GPRGNN | $0.6875 \pm 0.0038$ | $\mathbf{0.8443 \pm 0.0033}$ | $0.5101 \pm 0.0035$ | $\mathbf{0.4999 \pm 0.0025}$ |
| Chameleon | NLGCN | $0.5987 \pm 0.0068$ | $\mathbf{0.6657 \pm 0.0062}$ | $0.5233 \pm 0.0062$ | $\mathbf{0.4954 \pm 0.0065}$ |
| | NLGAT | $0.5926 \pm 0.0072$ | $\mathbf{0.6585 \pm 0.0070}$ | $0.5246 \pm 0.0065$ | $\mathbf{0.4902 \pm 0.0063}$ |
| | NLMLP | $0.4281 \pm 0.0078$ | $\mathbf{0.4824 \pm 0.0074}$ | $0.5623 \pm 0.0057$ | $\mathbf{0.4848 \pm 0.0051}$ |
| | GPRGNN | $0.5230 \pm 0.0054$ | $\mathbf{0.6550 \pm 0.0058}$ | $0.5107 \pm 0.0060$ | $\mathbf{0.4936 \pm 0.0049}$ |

Table 5: Performance comparison between TSD and DMP. Compared to DMP, TSD has an average increase in utility by 4.35%, with a comparable attack AUROC.

| Dataset | Models | Classify Acc (DMP) | Classify Acc (TSD) | Attack AUROC (DMP) | Attack AUROC (TSD) |
|---------|--------|--------------------|--------------------|--------------------|--------------------|
| PubMed | GCN | $0.8235 \pm 0.0037$ | $\mathbf{0.8387 \pm 0.0034}$ | $0.5026 \pm 0.0039$ | $\mathbf{0.4978 \pm 0.0042}$ |
| | GAT | $0.8027 \pm 0.0047$ | $\mathbf{0.8434 \pm 0.0024}$ | $0.5013 \pm 0.0036$ | $\mathbf{0.5005 \pm 0.0043}$ |
| | SGC | $0.8013 \pm 0.0041$ | $\mathbf{0.8096 \pm 0.0045}$ | $0.5024 \pm 0.0042$ | $\mathbf{0.5003 \pm 0.0038}$ |
| | GPRGNN | $0.8104 \pm 0.0031$ | $\mathbf{0.8423 \pm 0.0036}$ | $0.5020 \pm 0.0027$ | $\mathbf{0.4994 \pm 0.0023}$ |
| Facebook | GCN | $0.6896 \pm 0.0046$ | $\mathbf{0.7054 \pm 0.0045}$ | $0.4806 \pm 0.0032$ | $\mathbf{0.4964 \pm 0.0030}$ |
| | GAT | $0.6420 \pm 0.0040$ | $\mathbf{0.6797 \pm 0.0043}$ | $0.4768 \pm 0.0042$ | $\mathbf{0.4910 \pm 0.0037}$ |
| | SGC | $0.6152 \pm 0.0049$ | $\mathbf{0.6351 \pm 0.0040}$ | $0.4821 \pm 0.0038$ | $\mathbf{0.4969 \pm 0.0039}$ |
| | GPRGNN | $0.5784 \pm 0.0051$ | $\mathbf{0.5920 \pm 0.0046}$ | $0.4780 \pm 0.0045$ | $\mathbf{0.4964 \pm 0.0032}$ |
| Lastfm | GCN | $0.8162 \pm 0.0057$ | $\mathbf{0.8401 \pm 0.0055}$ | $0.5115 \pm 0.0030$ | $\mathbf{0.4978 \pm 0.0027}$ |
| | GAT | $0.8434 \pm 0.0054$ | $\mathbf{0.8769 \pm 0.0041}$ | $0.5142 \pm 0.0028$ | $\mathbf{0.4972 \pm 0.0023}$ |
| | SGC | $0.8112 \pm 0.0050$ | $\mathbf{0.8414 \pm 0.0044}$ | $0.5091 \pm 0.0036$ | $\mathbf{0.4967 \pm 0.0031}$ |
| | GPRGNN | $0.8140 \pm 0.0042$ | $\mathbf{0.8485 \pm 0.0037}$ | $0.5119 \pm 0.0026$ | $\mathbf{0.4979 \pm 0.0019}$ |
| Chameleon | NLGCN | $0.6681 \pm 0.0064$ | $\mathbf{0.6963 \pm 0.0065}$ | $0.5210 \pm 0.0064$ | $\mathbf{0.5182 \pm 0.0062}$ |
| | NLGAT | $0.6516 \pm 0.0066$ | $\mathbf{0.7082 \pm 0.0073}$ | $\mathbf{0.5116 \pm 0.0061}$ | $0.5159 \pm 0.0059$ |
| | NLMLP | $0.4643 \pm 0.0079$ | $\mathbf{0.4955 \pm 0.0070}$ | $0.5054 \pm 0.0053$ | $\mathbf{0.5017 \pm 0.0056}$ |
| | GPRGNN | $0.6471 \pm 0.0060$ | $\mathbf{0.6934 \pm 0.0068}$ | $0.5172 \pm 0.0067$ | $\mathbf{0.5163 \pm 0.0045}$ |

## 7 CONCLUSIONS AND LIMITATIONS

We proposed a novel two-stage defense method (TSD) against SMIAs for GNNs, and leveraged multiset functions to enhance SMIA attacks (End2end-SMIA). Our evaluation showed that TSD surpasses SHAN in defending against various types of SMIAs, establishing a new state-of-the-art benchmark. Additionally, we compared TSD with LBP and DMP in defending against classical node-level MIAs, demonstrating that TSD also performs effectively in protecting against the normal attacks. We conducted ablation studies and validated the origin of TSD's defense capability. TSD exhibits superior performance and is easy to integrate into various GNNs training processes.

**Limitations and Future Work.** The current form of TSD has the following limitations: (1) It can lead to lower model utility because the labels used in second stage are pseudolabels of test nodes, instead of groud-truth labels; (2) The flattening parameter $\beta$ is not end-to-end learnable, and uniform flattening may not be the optimal way to counter SMIAs and MIAs. To address these limitations, we are exploring to use only the test nodes with high confidence predictions and change the formula of soft labels to make $\beta$ learnable.

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

## A EXTENDED RELATED WORKS

**Membership Inference Attacks.** MIA on ML models aim to infer whether a data record was used to train a target ML model or not. This concept is firstly proposed by Homer et al. (2008) and later on extended to various directions, ranging from white-box setting where the whole target model is released (Nasr et al., 2019; Rezaei & Liu, 2021; Melis et al., 2019; Leino & Fredrikson, 2020), to black-box setting where only (partial of) output predictions are accessible to the adversary (Shokri et al., 2017; Salem et al., 2018; Song & Mittal, 2020; Li & Zhang, 2021; Choquette-Choo et al., 2021; Carlini et al., 2022). As a general guideline for MIA, the attacker first need to determine the most informative features that distinguish the sample membership. This feature can be posterior predictions (Shokri et al., 2017; Salem et al., 2018; Jia et al., 2019), loss values (Yeom et al., 2018; Sablayrolles et al., 2019), or gradient norms (Nasr et al., 2019; Rezaei & Liu, 2021). Upon identifying the informative features, the attacker can choose to learn either a binary classifier (Shokri et al., 2017) or metric-based decisions (Yeom et al., 2018; Salem et al., 2018) from shadow model trained on shadow dataset to extract patterns in these features among the training samples for identifying membership. The shadow dataset can be either generated from target model inferences, or a noisy version of the original dataset depending on the assumptions of the attacker.

**Defense Against Membership Inference Attacks.** As MIA exploit the behavioral differences of the target model on trainset and testset, most defense mechanisms work towards suppressing the common patterns that an optimal attack relies on. Popular defense methods include confidence score masking, regularization, knowledge distillation, and differential privacy. Confidence score masking aims to hide the true prediction vector returned by the target model and thus mitigates the effectiveness of MIAs, including only providing top-$k$ logits per inference (Shokri et al., 2017), or add noise to the prediction vector in an adversarial manner (Jia et al., 2019). Regularization aims to reduce the overfitting degree of target models to mitigate MIAs. Existing regularization methods including $L_2$-norm regularization (Choquette-Choo et al., 2021; Hayes et al., 2017), dropout (Leino & Fredrikson, 2020; Salem et al., 2018), data argumentation (Kaya & Dumitras, 2021; Yu et al., 2021), model compression (Wang et al., 2020), and label smoothing (Chen et al., 2022). Knowledge distillation aims to transfer the knowledge from a unprotected model to a protected model (Shejwalkar & Houmansadr, 2020), and differential privacy (Saeidian et al., 2021) naturally protects the membership information with theoretical guarantees at the cost of lower model utility.

## B STANDARD SMIA PROCESS

The Standard SMIA contains three phases: shadow GNN model training, attack model training, and membership inference. (1) shadow GNN model training: The adversary trains a set of shadow models $\Phi_1^s$ to $\Phi_T^s$ on shadow graphs to mimic the behaviors of the target model. In fact, training a single shadow model is sufficient to achieve performance comparable to training $T$ models. To train $\Phi_1^s$ to $\Phi_T^s$, the attackers will randomly sample the training dataset $\mathcal{G}_i^s$ for each shadow model from shadow dataset $\mathcal{G}_s$. Then the attackers train $\Phi_1^s$ to $\Phi_T^s$ by using $(X_i^s, A_i^s, Y_i^{\text{Train, s}}, \mathcal{V}_i^{\text{Train, s}})$. (2) attack model training: The attackers firstly generate a training dataset $\mathcal{A}^{\text{Train}}$ from the output of the shadow models on the shadow graph. The generation of $\mathcal{A}^{\text{Train}}$ follows four steps. First, for each shadow graph sample $\mathcal{G}_i^s$, the attackers select a set of $k$-node sets $\mathcal{V}_{att}^i$. Each node set $\mathcal{V}_{att} \in \mathcal{V}_{att}^i$ comprises $k$ nodes randomly selected from $\mathcal{G}_i^s$. Second, for each node set $\mathcal{V}_{att} \in \mathcal{V}_{att}^i$, the attackers obtain the posterior probability vector for each node in $\mathcal{V}_{att}$ output by the shadow model $\Phi_i^s$, so there will be $k$ posterior probability vectors for $\mathcal{V}_{att}$. Third, these $k$ posterior probability vectors are aggregated into a single vector which will serve as the attack feature vector $x$. Specifically, the attackers measure the pairwise similarity of the $k$ posterior probability vectors and obtains $\binom{k}{2}$ pairwise similarity values accordingly. Next, the attackers sort these similarity values in ascending order, and concatenates the sorted values as a vector. Following the the same choice in Wang & Wang (2024), we consider three similarity metrics, namely, dot product, cosine similarity, and euclidean distance. Therefore, there will be 3 sorted vectors accordingly, where each vector corresponds to a similarity metric. These 3 vectors are further concatenated as one vector, acting as the attack feature $x$. After the attackers generate the feature $x$ of the node set $\mathcal{V}_{att}$, they associates $x$ with its label $y$. In particular, $y = 1$ if $\mathcal{V}_{att}$ forms a $k$-clique in the $\mathcal{G}_i^s$, $y = 2$ if $\mathcal{V}_{att}$ contains a $(k-1)$-hop path, and $y = 0$ otherwise. Finally, the attackers add the newly formed data sample $(x, y)$ to $\mathcal{A}^{\text{Train}}$. After $\mathcal{A}^{\text{Train}}$ is generated, the attackers proceed to train the attack classifier $A$ on $\mathcal{A}^{\text{Train}}$. (3) membership inference: The attackers

employ the same methodology as the generation of training dataset $\mathcal{A}^{\text{Train}}$ to derive the feature $x_a$ for the target node set $\mathcal{V}_a$, utilizing the same similarity functions. It is important to note that, unlike the attack features of the training data $\mathcal{V}_{att}$ that uses the probability output by the shadow model, the attackers employ the posterior probability output of $\mathcal{V}_a$ by the target model to calculate $x_a$. Finally, the attackers feed $x_a$ into $A$ to obtain predictions. The complete standard SMIA process can be found in Wang & Wang (2024).

## C    STANDARD MIA PROCESS

The standard MIA process also has three phases: shadow GNN model training, attack model training, and membership inference. (1) shadow GNN model training: shadow GNN model $S$ is a model trained by attackers to replicate the behavior of the target GNN model $M$, providing training data for the attack model $A$. To train $S$, we assume that the shadow dataset $\mathcal{G}_s$ comes from the same or similar underlying distribution as $\mathcal{G}_t$. Then the attackers train $S$ by using $(X_s, A_s, Y_s^{\text{Train}}, \mathcal{V}_s^{\text{Train}})$ (2) attack model training: To train $A$, attackers use the trained $S$ to predict all nodes in $\mathcal{V}_s^{\text{Train}}$ and $\mathcal{V}_s^{\text{Test}}$ and obtain the corresponding posteriors. For each node, attackers take its posteriors as input of the attack model and assigns a label "1" if the node is from $\mathcal{V}_s^{\text{Train}}$ and "0" if the node is from $\mathcal{V}_s^{\text{Test}}$ to supervise. (3) membership inference: To implement membership inference attack on a given node $v$, attackers query $M$ with $v$'s feature to obtain its posterior. Then attackers input the posterior into the attack model to obtain the membership information. The complete standard MIA process can be found in Olatunji et al. (2021).

## D    COMPARISON WITH LBP AND DMP

Compared to state-of-the-art defense methods based on perturbations and distillation, such as LBP (Olatunji et al., 2021) for GNNs and DMP (Shejwalkar & Houmansadr, 2020) for graphless models, TSD can achieve a better balance between model utility and defense performance. LBP employs noise addition to the posteriors of the target model, grouping the elements randomly and adding noise from the same Laplace distribution to each group to reduce the required amount of noise. While LBP offers strong defense capabilities, the added noise significantly degrades the target model utility. DMP, on the other hand, tunes the data used for knowledge transfer to enhance membership privacy. It utilizes an unprotected model trained on private data to guide the training of a protected target model on reference data, optimizing the tradeoff between membership privacy and utility. However, DMP necessitates the collection of an additional dataset for training the protected model, which complicates the whole process. Meanwhile, our method addresses the overfitting problem implicitly by ensuring that both the training set and testset undergo the same procedure. Consequently, our method offers several advantages over LBP and DMP: (1) it avoids explicitly adding noise to the target model predictions, thereby preserving model utility; (2) it does not require additional data; and (3) it fully leverages the whole graph data through a train-test alternate training schedule.

## E    DETAILS OF DMP LOSS FUNCTION

In our experiments, the post-distillation phase of DMP consists of two parts of loss to train the protected model, with the proportion adjusted by a hyperparameter. One loss is the cross-entropy loss, supervised by the true labels of the reference data. The other loss is the KL divergence between the prediction of the protected model and the unprotected model on the reference data. The former is to ensure that the protected model has a high classification accuracy on the testset, while the latter is to guide the protected model by using the knowledge from the unprotected model. In our experiments, we adjust the hyperparameters to balance the testset classification accuracy and defense capability of the protected model.

## F    COMPLETE EXPERIMENTAL SETTINGS

Appendix Table 6 contains properties and statistics about benchmark datasets we used in SMIA experiments. For target models and shadow models, we used 2-layer GCN, 2-layer GAT, 2-layer

SGC, NLGCN, NLGAT, NLMLP, and GPRGNN architecture for CiteSeer, lastFM, Chameleon and 3-layer GCN, 3-layer GAT, 3-layer SGC, GPRGNN for Facebook. The attack model is a 3-layer MLP model. The optimizer we used is Adam. All target and shadow models are trained such that they achieve comparable performance as reported by the authors in the literature. We used one NVIDIA GeForce RTX 3090 for training. The time for finishing one experiment is about 10 minutes to 30 minutes depends on the complexity of datasets.

Table 6: Benchmark dataset properties and statistics of SMIA experiments: $|\mathcal{V}|$ and $|\mathcal{E}|$ denote the number of vertices and edges in the corresponding graph dataset.

| Dataset | $|\mathcal{V}|$ | $|\mathcal{E}|$ | Features | Classes |
|---|---|---|---|---|
| CiteSeer | 3327 | 4552 | 3703 | 6 |
| Facebook | 4039 | 88234 | 1283 | 193 |
| LastFM | 7624 | 55612 | 128 | 18 |
| Chameleon | 2277 | 31371 | 2325 | 5 |

Table 7: Benchmark dataset properties and statistics of MIA experiments: $|\mathcal{V}|$ and $|\mathcal{E}|$ denote the number of vertices and edges in the corresponding graph dataset.

| Dataset | $|\mathcal{V}|$ | $|\mathcal{E}|$ | Features | Classes |
|---|---|---|---|---|
| PubMed | 19717 | 44324 | 500 | 5 |
| Computers | 13752 | 245861 | 767 | 10 |
| Photo | 7650 | 119081 | 745 | 8 |
| Ogbn-Arxiv | 169343 | 1166243 | 128 | 40 |
| Texas | 183 | 279 | 1703 | 5 |
| Squirrel | 5201 | 198353 | 2089 | 5 |

Appendix Table 7 provides additional information on the properties and statistics of the datasets used in the MIA experiments. For target models and shadow models, we used 2-layer GCN, 2-layer GAT, 2-layer SGC, NLGCN, NLGAT, NLMLP, and GPRGNN architecture for PubMed, Computers, Photo, lastFM, Texas, Chameleon, Squirrel and 3-layer GCN, 3-layer GAT, 3-layer SGC, GPRGNN for Facebook and Ogbn-Arxiv. The attack model is a 3-layer MLP model. The optimizer we used is Adam. All target and shadow models are trained such that they achieve comparable performance as reported by the authors in the literature. We used one NVIDIA GeForce RTX 3090 for training. The time for finishing one experiment is about 10 minutes to 5 hours depends on the complexity of datasets.

## G COMPLETE RESULTS OF SMIA DEFENSE COMPARISON AND SMIA ATTACK COMPARISON

Appendix Table 8 and 9 show the complete results including standard deviation.

## H ADAPTIVE SMIA ATTACK

Adaptive SMIA attack refers to scenarios where the attacker is aware that the target model employs the TSD defense method. We consider two scenarios of adaptive attack.

The first scenario of an adaptive attack occurs when the attacker knows that the target model utilizes the TSD defense method and is familiar with the target model's architecture. Additionally, the attacker coincidentally selects the same dataset to train a shadow model. However, they are completely unaware of how the training and testing sets are partitioned within TSD, making the thresholds used for Flattening and the test set in the two-stage training unknown. In this case, it is difficult for the attacker to adjust their attack strategy to better counter the TSD method, even if they

Table 8: 3-SMIA attack performance comparison between Standard-SMIA and End2end-SMIA & Defense performance comparison between SHNA and TSD. Compared to SHNA, TSD achieves a decrease in attack AUROC by $16.64\%$ against Standard-SMIA & $14.30\%$ against End2end-SMIA, and increases in utility performance by $10.05\%$ on average.

| Dataset | Models | CA(SHNA) | CA(TSD) | S-SMIA AU (SHNA) | S-SMIA AU (TSD) | E-SMIA AU(SHNA) | E-SMIA AU(TSD) |
|---|---|---|---|---|---|---|---|
| CiteSeer | GCN | $0.6887 \pm 0.0066$ | $0.7729 \pm 0.0080$ | $0.9157 \pm 0.0072$ | $0.7767 \pm 0.0080$ | $0.9684 \pm 0.0084$ | $0.8703 \pm 0.0075$ |
| | GAT | $0.7082 \pm 0.0063$ | $0.7548 \pm 0.0092$ | $0.8784 \pm 0.0084$ | $0.7145 \pm 0.0075$ | $0.9336 \pm 0.0093$ | $0.8519 \pm 0.0094$ |
| | SGC | $0.6982 \pm 0.0077$ | $0.7454 \pm 0.0084$ | $0.9015 \pm 0.0069$ | $0.7978 \pm 0.0082$ | $0.9862 \pm 0.0089$ | $0.8904 \pm 0.0103$ |
| | GPRGNN | $0.6736 \pm 0.0069$ | $0.7864 \pm 0.0079$ | $0.8222 \pm 0.0080$ | $0.5969 \pm 0.0066$ | $0.9006 \pm 0.0078$ | $0.7364 \pm 0.0084$ |
| Facebook | GCN | $0.6028 \pm 0.0076$ | $0.7095 \pm 0.0081$ | $0.7143 \pm 0.0061$ | $0.5463 \pm 0.0064$ | $0.7942 \pm 0.0086$ | $0.6127 \pm 0.0096$ |
| | GAT | $0.5321 \pm 0.0073$ | $0.6435 \pm 0.0098$ | $0.6854 \pm 0.0057$ | $0.5382 \pm 0.0066$ | $0.8229 \pm 0.0085$ | $0.6068 \pm 0.0084$ |
| | SGC | $0.5011 \pm 0.0075$ | $0.6423 \pm 0.0091$ | $0.7028 \pm 0.0052$ | $0.5253 \pm 0.0058$ | $0.7903 \pm 0.0076$ | $0.5945 \pm 0.0129$ |
| | GPRGNN | $0.5529 \pm 0.0070$ | $0.6544 \pm 0.0079$ | $0.6512 \pm 0.0069$ | $0.5562 \pm 0.0072$ | $0.8081 \pm 0.0079$ | $0.6213 \pm 0.0072$ |
| LastFM | GCN | $0.8256 \pm 0.0072$ | $0.8523 \pm 0.0086$ | $0.9215 \pm 0.0088$ | $0.8337 \pm 0.0068$ | $0.9825 \pm 0.0078$ | $0.8702 \pm 0.0088$ |
| | GAT | $0.8379 \pm 0.0067$ | $0.8662 \pm 0.0077$ | $0.8528 \pm 0.0072$ | $0.8664 \pm 0.0065$ | $0.9274 \pm 0.0068$ | $0.8821 \pm 0.0075$ |
| | SGC | $0.8154 \pm 0.0064$ | $0.8400 \pm 0.0094$ | $0.9305 \pm 0.0089$ | $0.8835 \pm 0.0065$ | $0.9840 \pm 0.0087$ | $0.9060 \pm 0.0109$ |
| | GPRGNN | $0.8320 \pm 0.0071$ | $0.8577 \pm 0.0078$ | $0.9004 \pm 0.0093$ | $0.8268 \pm 0.0073$ | $0.9543 \pm 0.0074$ | $0.8687 \pm 0.0069$ |
| Chameleon | NLGCN | $0.6373 \pm 0.0071$ | $0.6725 \pm 0.0091$ | $0.8904 \pm 0.0076$ | $0.4538 \pm 0.0083$ | $0.9196 \pm 0.0074$ | $0.7875 \pm 0.0098$ |
| | NLGAT | $0.6437 \pm 0.0083$ | $0.6835 \pm 0.0078$ | $0.7025 \pm 0.0077$ | $0.5611 \pm 0.0079$ | $0.8848 \pm 0.0082$ | $0.7958 \pm 0.0074$ |
| | NLMLP | $0.5010 \pm 0.0069$ | $0.5484 \pm 0.0096$ | $0.7128 \pm 0.0096$ | $0.5342 \pm 0.0091$ | $0.8152 \pm 0.0091$ | $0.7057 \pm 0.0130$ |
| | GPRGNN | $0.6637 \pm 0.0059$ | $0.6835 \pm 0.0085$ | $0.7191 \pm 0.0087$ | $0.6259 \pm 0.0069$ | $0.8929 \pm 0.0092$ | $0.7711 \pm 0.0097$ |
| DBLP | GCN | $0.6201 \pm 0.0131$ | $0.7652 \pm 0.0117$ | $0.5462 \pm 0.0153$ | $0.5312 \pm 0.0134$ | $0.7124 \pm 0.0140$ | $0.6544 \pm 0.0128$ |
| | GAT | $0.6146 \pm 0.0145$ | $0.7721 \pm 0.0135$ | $0.5631 \pm 0.0114$ | $0.5116 \pm 0.0137$ | $0.6921 \pm 0.0124$ | $0.5974 \pm 0.0134$ |
| | SAGE | $0.6254 \pm 0.0116$ | $0.7486 \pm 0.0131$ | $0.5398 \pm 0.0151$ | $0.5267 \pm 0.0115$ | $0.6691 \pm 0.0134$ | $0.6235 \pm 0.0122$ |
| IMDB | GCN | $0.4366 \pm 0.0125$ | $0.5167 \pm 0.0131$ | $0.5132 \pm 0.0134$ | $0.5107 \pm 0.0120$ | $0.6518 \pm 0.0153$ | $0.5811 \pm 0.0113$ |
| | GAT | $0.4297 \pm 0.0143$ | $0.5283 \pm 0.0121$ | $0.5094 \pm 0.0137$ | $0.5026 \pm 0.0135$ | $0.6324 \pm 0.0129$ | $0.5637 \pm 0.0111$ |
| | SAGE | $0.4172 \pm 0.0134$ | $0.5244 \pm 0.0135$ | $0.5068 \pm 0.0150$ | $0.5024 \pm 0.0127$ | $0.6063 \pm 0.0128$ | $0.5644 \pm 0.0124$ |

Table 9: 4-SMIA attack performance comparison between Standard-SMIA and End2end-SMIA & Defense performance comparison between SHNA and TSD. Compared to SHNA, TSD achieves a decrease in attack AUROC by $16.56\%$ against Standard-SMIA & $13.89\%$ against End2end-SMIA, and increases in utility performance by $10.05\%$ on average.

| Dataset | Models | CA(SHNA) | CA(TSD) | S-SMIA AU(SHNA) | S-SMIA AU(TSD) | E-SMIA AU(SHNA) | E-SMIA AU(TSD) |
|---|---|---|---|---|---|---|---|
| CiteSeer | GCN | $0.6887 \pm 0.0066$ | $0.7729 \pm 0.0080$ | $0.9685 \pm 0.0079$ | $0.9013 \pm 0.0076$ | $0.9786 \pm 0.0081$ | $0.9267 \pm 0.0088$ |
| | GAT | $0.7082 \pm 0.0063$ | $0.7548 \pm 0.0092$ | $0.9718 \pm 0.0080$ | $0.8868 \pm 0.0079$ | $0.9891 \pm 0.0073$ | $0.9191 \pm 0.0079$ |
| | SGC | $0.6982 \pm 0.0077$ | $0.7454 \pm 0.0084$ | $0.9839 \pm 0.0082$ | $0.9316 \pm 0.0080$ | $0.9955 \pm 0.0068$ | $0.9442 \pm 0.0074$ |
| | GPRGNN | $0.6736 \pm 0.0069$ | $0.7864 \pm 0.0079$ | $0.9010 \pm 0.0077$ | $0.8523 \pm 0.0072$ | $0.9692 \pm 0.0091$ | $0.8894 \pm 0.0094$ |
| Facebook | GCN | $0.6028 \pm 0.0076$ | $0.7095 \pm 0.0081$ | $0.7377 \pm 0.0080$ | $0.5308 \pm 0.0082$ | $0.7964 \pm 0.0069$ | $0.6033 \pm 0.0074$ |
| | GAT | $0.5321 \pm 0.0073$ | $0.6435 \pm 0.0098$ | $0.6732 \pm 0.0078$ | $0.5027 \pm 0.0074$ | $0.7436 \pm 0.0085$ | $0.5978 \pm 0.0077$ |
| | SGC | $0.5011 \pm 0.0075$ | $0.6423 \pm 0.0091$ | $0.6992 \pm 0.0086$ | $0.5097 \pm 0.0089$ | $0.7520 \pm 0.0073$ | $0.6325 \pm 0.0073$ |
| | GPRGNN | $0.5529 \pm 0.0070$ | $0.6544 \pm 0.0079$ | $0.6520 \pm 0.0075$ | $0.4978 \pm 0.0078$ | $0.7090 \pm 0.0076$ | $0.5885 \pm 0.0089$ |
| LastFM | GCN | $0.8256 \pm 0.0072$ | $0.8523 \pm 0.0086$ | $0.9649 \pm 0.0094$ | $0.8296 \pm 0.0071$ | $0.9972 \pm 0.0094$ | $0.8561 \pm 0.0086$ |
| | GAT | $0.8379 \pm 0.0067$ | $0.8662 \pm 0.0077$ | $0.8936 \pm 0.0086$ | $0.7442 \pm 0.0082$ | $0.9354 \pm 0.0087$ | $0.7654 \pm 0.0080$ |
| | SGC | $0.8154 \pm 0.0064$ | $0.8401 \pm 0.0094$ | $0.9574 \pm 0.0079$ | $0.8184 \pm 0.0070$ | $0.9979 \pm 0.0092$ | $0.8270 \pm 0.0081$ |
| | GPRGNN | $0.8320 \pm 0.0071$ | $0.8577 \pm 0.0078$ | $0.9144 \pm 0.0089$ | $0.8044 \pm 0.0094$ | $0.9844 \pm 0.0076$ | $0.8444 \pm 0.0097$ |
| Chameleon | NLGCN | $0.6373 \pm 0.0071$ | $0.6725 \pm 0.0091$ | $0.7663 \pm 0.0074$ | $0.5668 \pm 0.0079$ | $0.9952 \pm 0.0086$ | $0.8346 \pm 0.0079$ |
| | NLGAT | $0.6437 \pm 0.0083$ | $0.6835 \pm 0.0078$ | $0.7849 \pm 0.0080$ | $0.5901 \pm 0.0087$ | $0.9816 \pm 0.0091$ | $0.8409 \pm 0.0063$ |
| | NLMLP | $0.5010 \pm 0.0069$ | $0.5484 \pm 0.0096$ | $0.7531 \pm 0.0070$ | $0.6076 \pm 0.0082$ | $0.9186 \pm 0.0076$ | $0.8359 \pm 0.0078$ |
| | GPRGNN | $0.6637 \pm 0.0059$ | $0.6835 \pm 0.0085$ | $0.8358 \pm 0.0075$ | $0.5924 \pm 0.0089$ | $0.9486 \pm 0.0094$ | $0.8127 \pm 0.0088$ |

are aware that the defense mechanism is TSD. If the attacker attempts to better simulate the target model by training the shadow model using the same approach as TSD, any significant differences in the partitioning of training and testing sets between the shadow and target models would actually weaken the effectiveness of the attack. Under such circumstances, the best strategy for the attacker is to train the shadow model in a standard manner without applying TSD.

The second scenario occurs when the attacker, in addition to the first case, also knows the target model's method for partitioning the dataset (though they still do not know the actual dataset used by the target model and have merely coincidentally chosen the same one). In this situation, if the attacker trains the shadow model in a standard manner, the results will be consistent with those in Tables 1 and 2, since the experimental setup in Section 6.1.1 aligns with the second scenario of the adaptive attack. However, in the case where the attacker employs the same training strategy as TSD, the attack will become stronger. Therefore, we conducted experiments under these circumstances to better test TSD defense method, and the experimental results are presented in Table10.

In experiments, we used both Standard-SMIA and End2end-SMIA. The experimental results indicate that compared to standard attacks, adaptive attacks indeed achieve higher AUROC and present

Table 10: The performance of TSD under Adaptive 3-SMIA attack. $no\ d$ indicates that no defense method was employed, $no\ a$ indicates that adaptive attacks were not used when the TSD defense was applied, $a$ indicates that adaptive attacks were employed when the TSD defense was applied.

| Dataset | Models | S-SMIA AU(no d) | S-SMIA AU (no a) | S-SMIA AU (a) | E-SMIA AU (no d) | E-SMIA AU (no a) | E-SMIA AU (a) |
|---------|--------|-----------------|------------------|---------------|------------------|------------------|---------------|
| CiteSeer | GCN | $0.9587 \pm 0.0078$ | $0.7767 \pm 0.0086$ | $0.8316 \pm 0.0074$ | $0.9821 \pm 0.0069$ | $0.8703 \pm 0.0091$ | $0.9245 \pm 0.0086$ |
| Facebook | GCN | $0.8362 \pm 0.0074$ | $0.5463 \pm 0.0079$ | $0.6581 \pm 0.0082$ | $0.9014 \pm 0.0077$ | $0.6127 \pm 0.0084$ | $0.7061 \pm 0.0081$ |
| LastFM | GCN | $0.9728 \pm 0.0076$ | $0.8337 \pm 0.0077$ | $0.8986 \pm 0.0078$ | $0.9931 \pm 0.0082$ | $0.8702 \pm 0.0094$ | $0.9211 \pm 0.0080$ |
| Chameleon | NLGCN | $0.9267 \pm 0.0073$ | $0.4538 \pm 0.0070$ | $0.6358 \pm 0.0081$ | $0.9747 \pm 0.0088$ | $0.7875 \pm 0.0085$ | $0.8553 \pm 0.0079$ |

greater challenges for defense mechanisms. However, when comparing the AUROC of attacks without any defense to that of adaptive attacks employing the TSD defense method, it is evident that TSD still maintains a significant defensive ability. Therefore, TSD is a highly effective defense method against SMIA attacks.

## I  COMPLETE ABLATION STUDY RESULTS OF TSD'S DEFENSE PERFORMANCE AGAINST SMIA

Appendix Table 11 and 12 shows the complete ablation study results of TSD's defense performance against SMIA. The experimental results show that the same conclusions as in Section 6.1.3 can be drawn across all datasets and GNN architectures. Additionally, it is worth noting the performance of MLP under Standard Training. MLP exhibits strong defense capabilities but poor classification performance. This aligns with intuition, as MLP does not use message passing and is therefore completely unaware of the structural information between nodes.

## J  COMPLETE EXPERIMENTS OF MIA DEFENSE

### J.1  COMPARISON WITH LBP

The parameters we used for LBP is shown in Appendix Table 13. For each experiment, we repeated 5 times and presented the mean and standard deviation of the results in Appendix Table 14.

Our analysis corresponding to different datasets is as follows: For PubMed, Computers, Photo, Facebook, LastFm and Ogbn-Arxiv, TSD achieves much better classify performance. However, there is a slight improvement in defense capability. This is because the average degree of nodes in these datasets is relatively large, and similar nodes tend to cluster in greater numbers. The target model can learn classification capabilities through a large number of similar node features, leading to more severe overfitting on the testsets, making the attack model more dangerous. Although LBP defense method can also achieve decent defense capability, it comes at the cost of significant loss in target model classification capability.

For Texas dataset, TSD shows significant improvements in both classification and defense capabilities. This is because the Texas dataset has a smaller number of nodes, leading to insufficient training data for the target model and severe overfitting. However, TSD converts the testset into training data for the target model, greatly enhancing the model's generalization ability and thus strengthening its defense capabilities. In contrast, the LBP defense method excessively sacrifices the model's classification ability, making it difficult to be utilized effectively.

For Chameleon and Squirrel dataset, it can be seen that even with very low model classification accuracy, the attack model can still achieve membership inference with a probability exceeding random selection. TSD demonstrates significant improvements in model classification on two datasets. We note that NLMLP's defense capability has been greatly enhanced, this is because the improvement in its generalization ability.

Table 11: Complete ablation study on the source of gains for TSD under 3-SMIA Attack.

| Method | Dataset | GNN Models | Classify Acc | Attack AUROC |
|---|---|---|---|---|
| Standard Training | CiteSeer | GCN | $0.8012 \pm 0.0093$ | $0.9745 \pm 0.0084$ |
| | | GAT | $0.7855 \pm 0.0086$ | $0.9628 \pm 0.0088$ |
| | | SGC | $0.7708 \pm 0.0094$ | $0.9743 \pm 0.0128$ |
| | | GPRGNN | $0.8247 \pm 0.0079$ | $0.8672 \pm 0.0086$ |
| | | MLP | $0.6988 \pm 0.0095$ | $0.6531 \pm 0.0098$ |
| | Facebook | GCN | $0.7353 \pm 0.0085$ | $0.8759 \pm 0.0091$ |
| | | GAT | $0.6745 \pm 0.0096$ | $0.8641 \pm 0.0094$ |
| | | SGC | $0.6721 \pm 0.0121$ | $0.8528 \pm 0.0105$ |
| | | GPRGNN | $0.7001 \pm 0.0094$ | $0.8740 \pm 0.0080$ |
| | | MLP | $0.3502 \pm 0.0079$ | $0.5746 \pm 0.0085$ |
| | LastFM | GCN | $0.8823 \pm 0.0088$ | $0.9841 \pm 0.0094$ |
| | | GAT | $0.8930 \pm 0.0094$ | $0.9854 \pm 0.0087$ |
| | | SGC | $0.8726 \pm 0.0093$ | $0.9832 \pm 0.0097$ |
| | | GPRGNN | $0.8991 \pm 0.0081$ | $0.9897 \pm 0.0073$ |
| | | MLP | $0.6808 \pm 0.0092$ | $0.6539 \pm 0.0082$ |
| | Chameleon | NLGCN | $0.7052 \pm 0.0085$ | $0.9664 \pm 0.0084$ |
| | | NLGAT | $0.7013 \pm 0.0097$ | $0.9748 \pm 0.0078$ |
| | | MLMLP | $0.5752 \pm 0.0115$ | $0.9086 \pm 0.0099$ |
| | | GPRGNN | $0.7054 \pm 0.0083$ | $0.9701 \pm 0.0093$ |
| | | MLP | $0.4569 \pm 0.0096$ | $0.5603 \pm 0.0087$ |
| Flattening (One-Stage) | CiteSeer | GCN | $0.7956 \pm 0.0082$ | $0.9428 \pm 0.0081$ |
| | | GAT | $0.7814 \pm 0.0074$ | $0.9331 \pm 0.0089$ |
| | | SGC | $0.7657 \pm 0.0098$ | $0.9452 \pm 0.0096$ |
| | | GPRGNN | $0.8194 \pm 0.0068$ | $0.8351 \pm 0.0072$ |
| | Facebook | GCN | $0.7291 \pm 0.0080$ | $0.8249 \pm 0.0084$ |
| | | GAT | $0.6684 \pm 0.0093$ | $0.8121 \pm 0.0096$ |
| | | SGC | $0.6667 \pm 0.0094$ | $0.7901 \pm 0.0091$ |
| | | GPRGNN | $0.6974 \pm 0.0083$ | $0.8298 \pm 0.0083$ |
| | LastFM | GCN | $0.8762 \pm 0.0096$ | $0.9520 \pm 0.0075$ |
| | | GAT | $0.8881 \pm 0.0091$ | $0.9461 \pm 0.0069$ |
| | | SGC | $0.8679 \pm 0.0126$ | $0.9487 \pm 0.0094$ |
| | | GPRGNN | $0.8924 \pm 0.0089$ | $0.9511 \pm 0.0080$ |
| | Chameleon | NLGCN | $0.7009 \pm 0.0094$ | $0.9187 \pm 0.0093$ |
| | | NLGAT | $0.6961 \pm 0.00101$ | $0.9258 \pm 0.0081$ |
| | | NLMLP | $0.5691 \pm 0.00134$ | $0.8609 \pm 0.0118$ |
| | | GPRGNN | $0.6977 \pm 0.0096$ | $0.9176 \pm 0.0090$ |

## J.2 COMPARISON WITH DMP

For each experiment, we repeated 5 times and presented the mean and standard deviation of the results in Appendix Table 15.

Our analysis corresponding to different datasets is as follows: For all datasets, compared to the DMP, TSD has a slight lead in both classification accuracy and defense performance. The knowledge distillation of the DMP method is pronounced in guiding the protected target model, and its defense capability is comparable to the TSD. However, the DMP method still results in a reduction in the amount of training data, which still has a significant negative impact on the model's classification ability.

In the experiments, we also observed that controlling the hyperparameters that determine the proportions of the two different losses in the post distillation phase of DMP is crucial. It requires achieving a tradeoff between classification accuracy and defense capability. Adjusting these hyperparameters will increase the implementation cost of the DMP method.

Table 12: Complete ablation study on the source of gains for TSD under 3-SMIA Attack. Continuation of Appendix Table 11.

| Method | Dataset | GNN Models | Classify Acc | Attack AUROC |
|---|---|---|---|---|
| Two-Stage (without Flattenning) | CiteSeer | GCN | $0.7804 \pm 0.0087$ | $0.9013 \pm 0.0081$ |
| | | GAT | $0.7627 \pm 0.0084$ | $0.8832 \pm 0.0086$ |
| | | SGC | $0.7529 \pm 0.0095$ | $0.9130 \pm 0.0090$ |
| | | GPRGNN | $0.7935 \pm 0.0078$ | $0.7628 \pm 0.0079$ |
| | Facebook | GCN | $0.7168 \pm 0.0080$ | $0.6644 \pm 0.0091$ |
| | | GAT | $0.6567 \pm 0.0086$ | $0.6621 \pm 0.0082$ |
| | | SGC | $0.6554 \pm 0.0094$ | $0.6510 \pm 0.0086$ |
| | | GPRGNN | $0.6639 \pm 0.0087$ | $0.6715 \pm 0.0084$ |
| | LastFM | GCN | $0.8610 \pm 0.0076$ | $0.9008 \pm 0.0091$ |
| | | GAT | $0.8749 \pm 0.0072$ | $0.9082 \pm 0.0087$ |
| | | SGC | $0.8532 \pm 0.0085$ | $0.9336 \pm 0.0104$ |
| | | GPRGNN | $0.8679 \pm 0.0081$ | $0.9017 \pm 0.0094$ |
| | Chameleon | NLGCN | $0.6863 \pm 0.0085$ | $0.8324 \pm 0.0074$ |
| | | NLGAT | $0.6944 \pm 0.0082$ | $0.8396 \pm 0.0082$ |
| | | NLMLP | $0.5571 \pm 0.0091$ | $0.7689 \pm 0.0118$ |
| | | GPRGNN | $0.6948 \pm 0.0071$ | $0.8302 \pm 0.0068$ |
| TSD (Two Stage & Flattenning) | CiteSeer | GCN | $0.7729 \pm 0.0080$ | $0.8703 \pm 0.0075$ |
| | | GAT | $0.7548 \pm 0.0092$ | $0.8519 \pm 0.0094$ |
| | | SGC | $0.7454 \pm 0.0084$ | $0.8904 \pm 0.0103$ |
| | | GPRGNN | $0.7864 \pm 0.0079$ | $0.7364 \pm 0.0084$ |
| | Facebook | GCN | $0.7095 \pm 0.0081$ | $0.6127 \pm 0.0096$ |
| | | GAT | $0.6435 \pm 0.0098$ | $0.6068 \pm 0.0084$ |
| | | SGC | $0.6423 \pm 0.0091$ | $0.5945 \pm 0.0129$ |
| | | GPRGNN | $0.6544 \pm 0.0079$ | $0.6213 \pm 0.0072$ |
| | LastFM | GCN | $0.8523 \pm 0.0086$ | $0.8702 \pm 0.0088$ |
| | | GAT | $0.8662 \pm 0.0077$ | $0.8821 \pm 0.0075$ |
| | | SGC | $0.8400 \pm 0.0094$ | $0.9060 \pm 0.0109$ |
| | | GPRGNN | $0.8577 \pm 0.0078$ | $0.8687 \pm 0.0069$ |
| | Chameleon | NLGCN | $0.6725 \pm 0.0091$ | $0.7875 \pm 0.0098$ |
| | | NLGAT | $0.6835 \pm 0.0078$ | $0.7958 \pm 0.0074$ |
| | | NLMLP | $0.5484 \pm 0.0096$ | $0.7057 \pm 0.0130$ |
| | | GPRGNN | $0.6835 \pm 0.0085$ | $0.7711 \pm 0.0097$ |

Table 13: Parameters for LBP

| Dataset | $N$ | $b$ |
|---|---|---|
| PubMed | 2 | 1 |
| Computers | 2 | 0.2 |
| Photo | 2 | 0.2 |
| Facebook | 2 | 0.2 |
| LastFM | 2 | 0.2 |
| Ogbn-Arxiv | 2 | 10 |
| Texas | 2 | 0.2 |
| Chameleon | 2 | 0.2 |
| Squirrel | 2 | 0.2 |

## J.3 TSD DEFENSE METHOD REDUCE THE GENERALIZATION GAP

In this section, we analyzed the changes in training loss and testing loss distribution before and after TSD training. Experiments are conducted on the heterophilic dataset Chameleon.

Through experiments, we demonstrated that TSD can: (1) reduce the gap between the average losses of training and testing nodes, thereby alleviating overfitting; (2) increase the variance of both

Table 14: Performance comparison between TSD and LBP. Compared to LBP, TSD achieves an increase in utility performance by $17.28\%$ on average, while achieving comparable attack AUROC to LBP.

| Dataset | Models | Classify Acc(LBP) | Classify Acc(TSD) | Attack AUROC(LBP) | Attack AUROC(TSD) |
|---|---|---|---|---|---|
| PubMed | GCN | $0.6886 \pm 0.0041$ | $0.8381 \pm 0.0023$ | $0.4998 \pm 0.0050$ | $0.4990 \pm 0.0048$ |
| | GAT | $0.7631 \pm 0.0037$ | $0.8400 \pm 0.0028$ | $0.5021 \pm 0.0084$ | $0.4911 \pm 0.0061$ |
| | SGC | $0.6564 \pm 0.0035$ | $0.8080 \pm 0.0020$ | $0.5007 \pm 0.0065$ | $0.5005 \pm 0.0057$ |
| | GPRGNN | $0.7843 \pm 0.0029$ | $0.8553 \pm 0.0014$ | $0.5003 \pm 0.0038$ | $0.4967 \pm 0.0034$ |
| Computers | GCN | $0.6900 \pm 0.0023$ | $0.8818 \pm 0.0018$ | $0.5128 \pm 0.0039$ | $0.5068 \pm 0.0035$ |
| | GAT | $0.7414 \pm 0.0026$ | $0.9086 \pm 0.0025$ | $0.5165 \pm 0.0053$ | $0.5039 \pm 0.0051$ |
| | SGC | $0.6383 \pm 0.0038$ | $0.8310 \pm 0.0023$ | $0.5112 \pm 0.0041$ | $0.5074 \pm 0.0043$ |
| | GPRGNN | $0.7203 \pm 0.0023$ | $0.8942 \pm 0.0012$ | $0.5154 \pm 0.0033$ | $0.5050 \pm 0.0030$ |
| Photo | GCN | $0.7737 \pm 0.0034$ | $0.9299 \pm 0.0023$ | $0.5179 \pm 0.0046$ | $0.5110 \pm 0.0041$ |
| | GAT | $0.8051 \pm 0.0040$ | $0.9453 \pm 0.0029$ | $0.5123 \pm 0.0045$ | $0.5066 \pm 0.0039$ |
| | SGC | $0.7361 \pm 0.0039$ | $0.9001 \pm 0.0028$ | $0.5159 \pm 0.0039$ | $0.5106 \pm 0.0035$ |
| | GPRGNN | $0.8187 \pm 0.0027$ | $0.9430 \pm 0.0015$ | $0.5153 \pm 0.0025$ | $0.5088 \pm 0.0028$ |
| Facebook | GCN | $0.5195 \pm 0.0041$ | $0.6778 \pm 0.0049$ | $0.4912 \pm 0.0021$ | $0.4993 \pm 0.0023$ |
| | GAT | $0.5460 \pm 0.0037$ | $0.6519 \pm 0.0039$ | $0.5120 \pm 0.0025$ | $0.5010 \pm 0.0028$ |
| | SGC | $0.4833 \pm 0.0044$ | $0.6249 \pm 0.0043$ | $0.4901 \pm 0.0026$ | $0.5004 \pm 0.0031$ |
| | GPRGNN | $0.4627 \pm 0.0032$ | $0.5890 \pm 0.0035$ | $0.4807 \pm 0.0031$ | $0.5014 \pm 0.0020$ |
| Lastfm | GCN | $0.6509 \pm 0.0037$ | $0.8378 \pm 0.0035$ | $0.5118 \pm 0.0024$ | $0.4971 \pm 0.0022$ |
| | GAT | $0.7210 \pm 0.0034$ | $0.8683 \pm 0.0032$ | $0.5136 \pm 0.0031$ | $0.4980 \pm 0.0029$ |
| | SGC | $0.6395 \pm 0.0040$ | $0.8336 \pm 0.0045$ | $0.5121 \pm 0.0030$ | $0.4965 \pm 0.0034$ |
| | GPRGNN | $0.6875 \pm 0.0038$ | $0.8443 \pm 0.0033$ | $0.5101 \pm 0.0035$ | $0.4999 \pm 0.0025$ |
| Ogbn-Arxiv | GCN | $0.5097 \pm 0.0025$ | $0.7200 \pm 0.0019$ | $0.5005 \pm 0.0037$ | $0.4995 \pm 0.0032$ |
| | GAT | $0.5134 \pm 0.0030$ | $0.7223 \pm 0.0024$ | $0.5001 \pm 0.0045$ | $0.4978 \pm 0.0038$ |
| | SGC | $0.5021 \pm 0.0032$ | $0.7272 \pm 0.0027$ | $0.4998 \pm 0.0040$ | $0.4923 \pm 0.0034$ |
| | GPRGNN | $0.5221 \pm 0.0021$ | $0.7315 \pm 0.0016$ | $0.5017 \pm 0.0031$ | $0.5002 \pm 0.0026$ |
| Texas | NLGCN | $0.5113 \pm 0.0033$ | $0.6152 \pm 0.0031$ | $0.5954 \pm 0.0061$ | $0.4812 \pm 0.0054$ |
| | NLGAT | $0.5327 \pm 0.0037$ | $0.5882 \pm 0.0026$ | $0.5710 \pm 0.0045$ | $0.4785 \pm 0.0043$ |
| | NLMLP | $0.5713 \pm 0.0062$ | $0.6686 \pm 0.0043$ | $0.5585 \pm 0.0035$ | $0.5532 \pm 0.0038$ |
| | GPRGNN | $0.6166 \pm 0.0041$ | $0.7224 \pm 0.0029$ | $0.6201 \pm 0.0038$ | $0.4559 \pm 0.0032$ |
| Chameleon | NLGCN | $0.5987 \pm 0.0068$ | $0.6657 \pm 0.0062$ | $0.5233 \pm 0.0062$ | $0.4954 \pm 0.0065$ |
| | NLGAT | $0.5926 \pm 0.0072$ | $0.6585 \pm 0.0070$ | $0.5246 \pm 0.0065$ | $0.4902 \pm 0.0063$ |
| | NLMLP | $0.4281 \pm 0.0078$ | $0.4824 \pm 0.0074$ | $0.5623 \pm 0.0057$ | $0.4848 \pm 0.0051$ |
| | GPRGNN | $0.5230 \pm 0.0054$ | $0.6550 \pm 0.0058$ | $0.5107 \pm 0.0060$ | $0.4936 \pm 0.0049$ |
| Squirrel | NLGCN | $0.4056 \pm 0.0078$ | $0.4910 \pm 0.0073$ | $0.5239 \pm 0.0059$ | $0.4913 \pm 0.0060$ |
| | NLGAT | $0.4503 \pm 0.0082$ | $0.5446 \pm 0.0077$ | $0.5260 \pm 0.0053$ | $0.4920 \pm 0.0049$ |
| | NLMLP | $0.2964 \pm 0.0085$ | $0.3137 \pm 0.0068$ | $0.5659 \pm 0.0058$ | $0.4746 \pm 0.0054$ |
| | GPRGNN | $0.3449 \pm 0.0066$ | $0.4013 \pm 0.0060$ | $0.5225 \pm 0.0055$ | $0.4922 \pm 0.0050$ |

member and non-member loss distributions and reduce the disparity between their means; and (3) decrease the distinguishability between member and non-member loss distributions.

**Reduce the gap between the average losses of training and testing nodes.** Appendix Figure 4 shows the variations of the average losses of training and testing nodes with increasing training epochs for both standard training and two-stage training on Chameleon dataset. We also recorded the losses of all models from Appendix Figure 4 at the end of training to Appendix Table 16, and additionally added the result of comparative experiments on model utility and defense capability. Comparing Appendix Figure 4 (a) with (b), it can be observed that the difference between the average losses of training and testing nodes in the standard training increases as epochs increase, indicating that overfitting exists and becomes worse as training proceeds. However, when using two-stage training, although overfitting cannot be completely avoided, the difference between training and testing losses decreases in the second stage as training proceeds, indicating a gradual alleviation of overfitting. Appendix Table 16 also shows that our method achieved lower average loss gap after the entire training process. All experimental results demonstrate the capability of our method to reduce overfitting and the generalization gap.

**Increase the variance of both member and non-member loss distributions and reduce the disparity between their means.** Appendix Figure 5 illustrates the loss distributions of member and

Table 15: Performance comparison between TSD and DMP. Compared to DMP, TSD achieves an increase in utility performance by $4.35\%$ on average, while achieving comparable attack AUROC to DMP.

| Dataset | Models | Classify Acc (DMP) | Classify Acc (TSD) | Attack AUROC (DMP) | Attack AUROC (TSD) |
|---|---|---|---|---|---|
| PubMed | GCN | $0.8235 \pm 0.0037$ | $0.8387 \pm 0.0034$ | $0.5026 \pm 0.0039$ | $0.4978 \pm 0.0042$ |
| | GAT | $0.8027 \pm 0.0047$ | $0.8434 \pm 0.0024$ | $0.5013 \pm 0.0036$ | $0.5005 \pm 0.0043$ |
| | SGC | $0.8013 \pm 0.0041$ | $0.8096 \pm 0.0045$ | $0.5024 \pm 0.0042$ | $0.5003 \pm 0.0038$ |
| | GPRGNN | $0.8104 \pm 0.0031$ | $0.8423 \pm 0.0036$ | $0.5020 \pm 0.0027$ | $0.4994 \pm 0.0023$ |
| Computers | GCN | $0.8756 \pm 0.0038$ | $0.8808 \pm 0.0040$ | $0.5055 \pm 0.0042$ | $0.5004 \pm 0.0045$ |
| | GAT | $0.9071 \pm 0.0044$ | $0.9127 \pm 0.0038$ | $0.5097 \pm 0.0045$ | $0.4933 \pm 0.0051$ |
| | SGC | $0.8323 \pm 0.0042$ | $0.8434 \pm 0.0040$ | $0.5086 \pm 0.0045$ | $0.5023 \pm 0.0047$ |
| | GPRGNN | $0.8683 \pm 0.0029$ | $0.8898 \pm 0.0021$ | $0.5045 \pm 0.0020$ | $0.5036 \pm 0.0032$ |
| Photo | GCN | $0.9228 \pm 0.0046$ | $0.9304 \pm 0.0042$ | $0.5061 \pm 0.0052$ | $0.5004 \pm 0.0048$ |
| | GAT | $0.9415 \pm 0.0052$ | $0.9493 \pm 0.0038$ | $0.5072 \pm 0.0058$ | $0.4966 \pm 0.0055$ |
| | SGC | $0.8933 \pm 0.0042$ | $0.8988 \pm 0.0037$ | $0.5105 \pm 0.0060$ | $0.5018 \pm 0.0059$ |
| | GPRGNN | $0.9215 \pm 0.0032$ | $0.9315 \pm 0.0026$ | $0.5043 \pm 0.0044$ | $0.4976 \pm 0.0042$ |
| Facebook | GCN | $0.6896 \pm 0.0046$ | $0.7054 \pm 0.0045$ | $0.4806 \pm 0.0032$ | $0.4964 \pm 0.0030$ |
| | GAT | $0.6420 \pm 0.0040$ | $0.6797 \pm 0.0043$ | $0.4768 \pm 0.0042$ | $0.4910 \pm 0.0037$ |
| | SGC | $0.6152 \pm 0.0049$ | $0.6351 \pm 0.0040$ | $0.4821 \pm 0.0038$ | $0.4969 \pm 0.0039$ |
| | GPRGNN | $0.5784 \pm 0.0051$ | $0.5920 \pm 0.0046$ | $0.4780 \pm 0.0045$ | $0.4964 \pm 0.0032$ |
| Lastfm | GCN | $0.8162 \pm 0.0057$ | $0.8401 \pm 0.0055$ | $0.5115 \pm 0.0030$ | $0.4978 \pm 0.0027$ |
| | GAT | $0.8434 \pm 0.0054$ | $0.8769 \pm 0.0041$ | $0.5142 \pm 0.0028$ | $0.4972 \pm 0.0023$ |
| | SGC | $0.8112 \pm 0.0050$ | $0.8414 \pm 0.0044$ | $0.5091 \pm 0.0036$ | $0.4967 \pm 0.0031$ |
| | GPRGNN | $0.8140 \pm 0.0042$ | $0.8485 \pm 0.0037$ | $0.5119 \pm 0.0026$ | $0.4979 \pm 0.0019$ |
| Ogbn-Arxiv | GCN | $0.6876 \pm 0.0039$ | $0.6921 \pm 0.0020$ | $0.4920 \pm 0.0045$ | $0.4952 \pm 0.0049$ |
| | GAT | $0.6869 \pm 0.0037$ | $0.6907 \pm 0.0018$ | $0.4913 \pm 0.0042$ | $0.4934 \pm 0.0048$ |
| | SGC | $0.6798 \pm 0.0023$ | $0.6884 \pm 0.0023$ | $0.4924 \pm 0.0048$ | $0.4991 \pm 0.0043$ |
| | GPRGNN | $0.6903 \pm 0.0025$ | $0.6993 \pm 0.0020$ | $0.5035 \pm 0.0037$ | $0.4972 \pm 0.0040$ |
| Texas | NLGCN | $0.6846 \pm 0.0069$ | $0.7027 \pm 0.0044$ | $0.4966 \pm 0.0060$ | $0.4970 \pm 0.0056$ |
| | NLGAT | $0.6916 \pm 0.0074$ | $0.7263 \pm 0.0038$ | $0.4923 \pm 0.0063$ | $0.4976 \pm 0.0058$ |
| | NLMLP | $0.6948 \pm 0.0070$ | $0.7282 \pm 0.0042$ | $0.4926 \pm 0.0068$ | $0.4953 \pm 0.0063$ |
| | GPRGNN | $0.6115 \pm 0.0054$ | $0.7445 \pm 0.0031$ | $0.5187 \pm 0.0052$ | $0.5032 \pm 0.0046$ |
| Chameleon | NLGCN | $0.6681 \pm 0.0064$ | $0.6963 \pm 0.0065$ | $0.5210 \pm 0.0064$ | $0.5182 \pm 0.0062$ |
| | NLGAT | $0.6516 \pm 0.0066$ | $0.7082 \pm 0.0073$ | $0.5116 \pm 0.0061$ | $0.5159 \pm 0.0059$ |
| | NLMLP | $0.4643 \pm 0.0079$ | $0.4955 \pm 0.0070$ | $0.5054 \pm 0.0053$ | $0.5017 \pm 0.0056$ |
| | GPRGNN | $0.6471 \pm 0.0060$ | $0.6934 \pm 0.0068$ | $0.5172 \pm 0.0067$ | $0.5163 \pm 0.0045$ |
| Squirrel | NLGCN | $0.5011 \pm 0.0074$ | $0.5102 \pm 0.0076$ | $0.5053 \pm 0.0064$ | $0.5001 \pm 0.0058$ |
| | NLGAT | $0.5502 \pm 0.0087$ | $0.5723 \pm 0.0071$ | $0.5146 \pm 0.0058$ | $0.4934 \pm 0.0055$ |
| | NLMLP | $0.3240 \pm 0.0090$ | $0.3393 \pm 0.0073$ | $0.5035 \pm 0.0055$ | $0.5098 \pm 0.0050$ |
| | GPRGNN | $0.4424 \pm 0.0061$ | $0.4581 \pm 0.0068$ | $0.5102 \pm 0.0051$ | $0.5054 \pm 0.0052$ |

Table 16: The performance comparison between standard training and TSD training.

| Dataset | Models | Avg Train Loss | | Avg Test Loss | | Classify Acc | | Attack AUROC | |
|---|---|---|---|---|---|---|---|---|---|
| | | Standard | TSD | Standard | TSD | Standard | TSD | Standard | TSD |
| Chameleon | NLGCN | 0.4988 | 0.6192 | 1.1278 | 1.2631 | 0.6631 | 0.6657 | 0.5267 | 0.4954 |
| | NLGAT | 0.1617 | 0.5782 | 1.1650 | 1.28911 | 0.6603 | 0.6585 | 0.5304 | 0.4902 |
| | NLMLP | 0.1555 | 0.9322 | 3.6396 | 2.0010 | 0.4857 | 0.4824 | 0.7681 | 0.4848 |
| | GPRGNN | 0.3575 | 0.8498 | 1.6416 | 1.3626 | 0.6582 | 0.6550 | 0.5837 | 0.4936 |

non-member nodes on the Chameleon dataset with NLGCN after training with both standard and two-stage methods. In the figure, members refer to the nodes in the trainset of the target model, while non-members refer to the nodes in the testset. Therefore, Figure 5 can also be viewed as the training and testing loss distributions of the target model after using different training methods. Comparing Figure 5 (a) with (b), it can be observed that the loss distributions of members and non-members after standard training have relatively small variances, and their means differ significantly. This conclusion is consistent with the results of the average losses in Appendix Table 16. However, after using two-stage training, significant changes occur in the loss distributions: the variances of both two distributions increase. And combined with the results in Appendix Table 16, it is obvious that their means become closer.

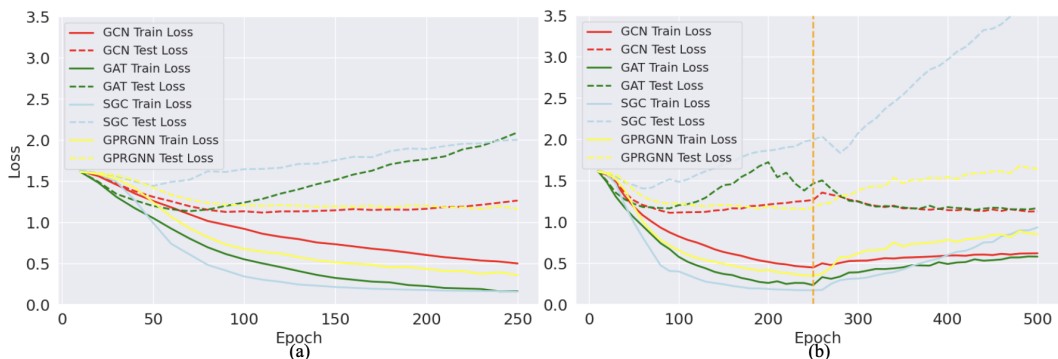

Figure 4: Comparison of average training loss and testing loss on Chameleon for (a) standard training, (b) two-stage training (TSD). In (b), the left half of the orange dashed line indicates the first training stage of our method, while the one on the right indicates the second stage.

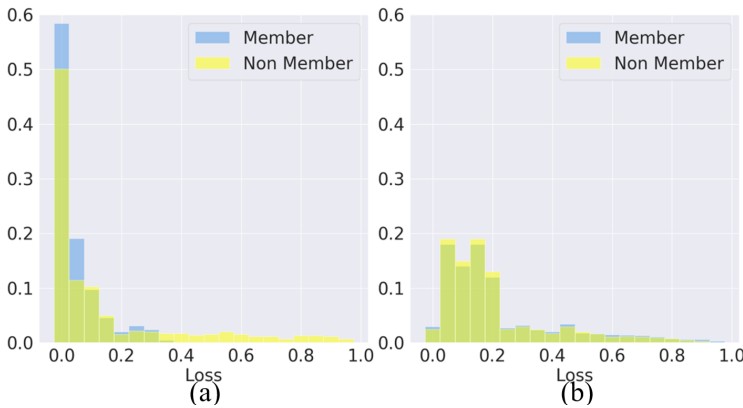

Figure 5: Loss distribution histograms for (a) standard training on Chameleon, (b) two-stage training (TSD) on Chameleon

**Decrease the distinguishability between member and non-member loss distributions.** From Figure 5, it can be seen that the overlap between the member and non-member loss distributions of the target model after two-stage training is significantly larger than that of standard training. Combined with the conclusions obtained above, we can confirm that the distinguishability between member and non-member distributions has decreased, which will increase the difficulty of MIA.

In summary, the changes of the target model induced by our two-stage training method are significant. Appendix Table 16 also demonstrates that such changes not only substantially enhance defense capability but also result in only subtle decline in downstream classification accuracy.

## J.4 DATA TRANSFER

Figure 6, 7 display the experimental results using GCN as backbones. The results demonstrate that TSD also exhibits excellent defense capability in the data transfer setting. Since model utility is only related to the target dataset, combining the classification performance of TSD in Appendix Table 14 and 15, it is evident that TSD can still achieve an outstanding balance between model utility and defense capability in the data transfer setting, which means that TSD can be effectively deployed in real-world applications.

## J.5 ABLATION STUDY OF TSD'S DEFENSE PERFORMANCE AGAINST MIA

In the experiments, we set up the same four variants as described in Section 6.1.3. Additionally, we also considered RelaxLoss (Chen et al., 2022), which is essentially a combination of alternate flattening and gradient ascent when the training loss falls below a predefined threshold. We use

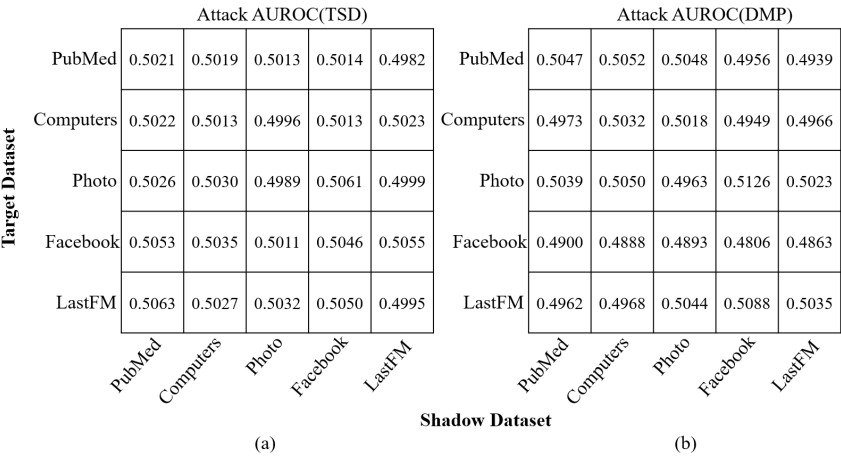

Figure 6: Comparison of TSD and LBP's defense performance under data transfer setting. (a) Attack AUROC of TSD; (b) Attack AUROC of LBP.

Figure 7: Comparison of TSD and DMP's defense performance under data transfer setting. (a) Attack AUROC of TSD; (b) Attack AUROC of DMP.

RelaxLoss as an example to show the difference between the defense methods effective for graph and graphless models, and necessities to design defense mechanisms specially for graph models. Note that the split ratio of trainsets and testsets for TSD is 9:1.

Appendix Table 17, 18,19 presents the results of ablation study regarding five variants. Our analysis is as follow: We first focus on the improvement of defense capability. It can be observed that two-stage (without flattening), flattening, and gradient ascent all enhance the defense capability of the target model compared to standard training. The effect of two-stage (without flattening) on reducing the AUROC of the attack model is the most pronounced, followed by flattening, while gradient ascent slightly reduces it. These results align with expectations because two-stage (without flattening) directly enables the model to learn the distribution of the testing data and flattening decrease the difference between the loss distributions' variances of training and testing nodes. Surprisingly, gradient ascent hardly improves the model's defense capability, suggesting that our method's exclusion of gradient ascent is reasonable.

Then we focus on the decline of classification accuracy caused by these variants. From the results, it can be seen that two-stage (without flattening) and gradient ascent hardly lead to a decrease in classification accuracy, and flattening only results in a slight decline. These results are interpretable: two-stage (without flattening) only used testing data for an extra training; gradient ascent has a minimal impact on the model's defense capability, which also means that it hardly change the model;

flattening slightly alters the model's mapping when using soft labels, so the classification accuracy decrease. However, the degree of decline caused by flattening is acceptable compared to its enhancement in defense capability. In summary, our TSD method (two stage with flattening) is the best.

Table 17: Ablation Study of TSD's Defense performance against MIA.

| Method | Dataset | GNN Models | Classify Acc | Attack AUROC |
|---|---|---|---|---|
| Standard Training | PubMed | GCN | $0.8423 \pm 0.0027$ | $0.5306 \pm 0.0032$ |
| | | GAT | $0.8421 \pm 0.0025$ | $0.5306 \pm 0.0043$ |
| | | SGC | $0.8129 \pm 0.0024$ | $0.5335 \pm 0.0047$ |
| | | GPRGNN | $0.8637 \pm 0.0026$ | $0.5388 \pm 0.0045$ |
| | Computers | GCN | $0.8865 \pm 0.0025$ | $0.5331 \pm 0.0032$ |
| | | GAT | $0.9145 \pm 0.0026$ | $0.5360 \pm 0.0043$ |
| | | SGC | $0.8419 \pm 0.0022$ | $0.5334 \pm 0.0041$ |
| | | GPRGNN | $0.8917 \pm 0.0023$ | $0.5377 \pm 0.0047$ |
| | Photo | GCN | $0.9349 \pm 0.0022$ | $0.5381 \pm 0.0030$ |
| | | GAT | $0.9471 \pm 0.0023$ | $0.5388 \pm 0.0039$ |
| | | SGC | $0.9122 \pm 0.0025$ | $0.5362 \pm 0.0042$ |
| | | GPRGNN | $0.9499 \pm 0.0028$ | $0.5390 \pm 0.0042$ |
| | Facebook | GCN | $0.6797 \pm 0.0034$ | $0.5352 \pm 0.0041$ |
| | | GAT | $0.6465 \pm 0.0038$ | $0.5426 \pm 0.0031$ |
| | | SGC | $0.6275 \pm 0.0042$ | $0.5341 \pm 0.0035$ |
| | | GPRGNN | $0.5635 \pm 0.0043$ | $0.5378 \pm 0.0039$ |
| | LastFM | GCN | $0.8355 \pm 0.0033$ | $0.5362 \pm 0.0036$ |
| | | GAT | $0.8692 \pm 0.0036$ | $0.5375 \pm 0.0041$ |
| | | SGC | $0.8334 \pm 0.0030$ | $0.5360 \pm 0.0045$ |
| | | GPRGNN | $0.8465 \pm 0.0029$ | $0.5400 \pm 0.0050$ |
| | Chameleon | NLGCN | $0.6817 \pm 0.0054$ | $0.5469 \pm 0.0051$ |
| | | NLGAT | $0.6451 \pm 0.0053$ | $0.5790 \pm 0.0058$ |
| | | NLMLP | $0.5077 \pm 0.0060$ | $0.7868 \pm 0.0049$ |
| | | GPRGNN | $0.6841 \pm 0.0057$ | $0.5939 \pm 0.0042$ |
| Flattenning (One-Stage) | PubMed | GCN | $0.8345 \pm 0.0025$ | $0.5278 \pm 0.0041$ |
| | | GAT | $0.8328 \pm 0.0023$ | $0.5213 \pm 0.0054$ |
| | | SGC | $0.8040 \pm 0.0023$ | $0.5263 \pm 0.0053$ |
| | | GPRGNN | $0.8541 \pm 0.0016$ | $0.5224 \pm 0.0036$ |
| | Computers | GCN | $0.8821 \pm 0.0026$ | $0.5254 \pm 0.0044$ |
| | | GAT | $0.9124 \pm 0.0023$ | $0.5236 \pm 0.0038$ |
| | | SGC | $0.8389 \pm 0.0028$ | $0.5267 \pm 0.0042$ |
| | | GPRGNN | $0.8892 \pm 0.0034$ | $0.5276 \pm 0.0039$ |
| | Photo | GCN | $0.9302 \pm 0.0032$ | $0.5297 \pm 0.0038$ |
| | | GAT | $0.9435 \pm 0.0035$ | $0.5268 \pm 0.0047$ |
| | | SGC | $0.9087 \pm 0.0035$ | $0.5275 \pm 0.0049$ |
| | | GPRGNN | $0.9468 \pm 0.0033$ | $0.5308 \pm 0.0046$ |
| | Facebook | GCN | $0.6751 \pm 0.0037$ | $0.5275 \pm 0.0039$ |
| | | GAT | $0.6458 \pm 0.0033$ | $0.5335 \pm 0.0034$ |
| | | SGC | $0.6236 \pm 0.0035$ | $0.5263 \pm 0.0042$ |
| | | GPRGNN | $0.5614 \pm 0.0040$ | $0.5267 \pm 0.0045$ |
| | LastFM | GCN | $0.8321 \pm 0.0037$ | $0.5245 \pm 0.0036$ |
| | | GAT | $0.8659 \pm 0.0039$ | $0.5255 \pm 0.0044$ |
| | | SGC | $0.8309 \pm 0.0036$ | $0.5282 \pm 0.0048$ |
| | | GPRGNN | $0.8430 \pm 0.0043$ | $0.5325 \pm 0.0046$ |
| | Chameleon | NLGCN | $0.6720 \pm 0.0068$ | $0.5327 \pm 0.0055$ |
| | | NLGAT | $0.6364 \pm 0.0066$ | $0.5685 \pm 0.0064$ |
| | | NLMLP | $0.4954 \pm 0.0075$ | $0.7787 \pm 0.0056$ |
| | | GPRGNN | $0.6757 \pm 0.0053$ | $0.5833 \pm 0.0039$ |

Table 18: Ablation Study of TSD's Defense performance against MIA. Continuation of Appendix Table 17.

| Method | Dataset | GNN Models | Classify Acc | Attack AUROC |
|---|---|---|---|---|
| Flattenning & Gradient Asent (One-Stage) | PubMed | GCN | $0.8310 \pm 0.0043$ | $0.5282 \pm 0.0050$ |
| | | GAT | $0.8430 \pm 0.0040$ | $0.5201 \pm 0.0054$ |
| | | SGC | $0.8190 \pm 0.0066$ | $0.5289 \pm 0.0050$ |
| | | GPRGNN | $0.8657 \pm 0.0041$ | $0.5242 \pm 0.0042$ |
| | Computers | GCN | $0.8798 \pm 0.0033$ | $0.5248 \pm 0.0045$ |
| | | GAT | $0.9084 \pm 0.0027$ | $0.5245 \pm 0.0044$ |
| | | SGC | $0.8378 \pm 0.0024$ | $0.5253 \pm 0.0047$ |
| | | GPRGNN | $0.8849 \pm 0.0027$ | $0.5257 \pm 0.0045$ |
| | Photo | GCN | $0.9268 \pm 0.0029$ | $0.5264 \pm 0.0047$ |
| | | GAT | $0.9389 \pm 0.0024$ | $0.5275 \pm 0.0045$ |
| | | SGC | $0.9057 \pm 0.0026$ | $0.5245 \pm 0.0043$ |
| | | GPRGNN | $0.9399 \pm 0.0025$ | $0.5288 \pm 0.0048$ |
| | Facebook | GCN | $0.6732 \pm 0.0025$ | $0.5243 \pm 0.0044$ |
| | | GAT | $0.6363 \pm 0.0024$ | $0.5342 \pm 0.0047$ |
| | | SGC | $0.6180 \pm 0.0028$ | $0.5251 \pm 0.0048$ |
| | | GPRGNN | $0.5591 \pm 0.0030$ | $0.5278 \pm 0.0046$ |
| | LastFM | GCN | $0.8284 \pm 0.0026$ | $0.5264 \pm 0.0042$ |
| | | GAT | $0.8592 \pm 0.0028$ | $0.5235 \pm 0.0041$ |
| | | SGC | $0.8284 \pm 0.0024$ | $0.5264 \pm 0.0046$ |
| | | GPRGNN | $0.8372 \pm 0.0027$ | $0.5342 \pm 0.0049$ |
| | Chameleon | NLGCN | $0.6707 \pm 0.0068$ | $0.5302 \pm 0.0074$ |
| | | NLGAT | $0.6402 \pm 0.0074$ | $0.5561 \pm 0.0077$ |
| | | NLMLP | $0.4933 \pm 0.0075$ | $0.7741 \pm 0.0063$ |
| | | GPRGNN | $0.6753 \pm 0.0064$ | $0.5829 \pm 0.0062$ |
| Two-Stage (without Flattening) | PubMed | GCN | $0.8328 \pm 0.0035$ | $0.5049 \pm 0.0043$ |
| | | GAT | $0.8489 \pm 0.0031$ | $0.5041 \pm 0.0070$ |
| | | SGC | $0.8055 \pm 0.0028$ | $0.5038 \pm 0.0063$ |
| | | GPRGNN | $0.8685 \pm 0.0023$ | $0.5044 \pm 0.0035$ |
| | Computers | GCN | $0.8868 \pm 0.0033$ | $0.5026 \pm 0.0045$ |
| | | GAT | $0.9149 \pm 0.0026$ | $0.5035 \pm 0.0043$ |
| | | SGC | $0.8393 \pm 0.0031$ | $0.5036 \pm 0.0039$ |
| | | GPRGNN | $0.8921 \pm 0.0023$ | $0.5045 \pm 0.0047$ |
| | Photo | GCN | $0.9326 \pm 0.0028$ | $0.5046 \pm 0.0042$ |
| | | GAT | $0.9464 \pm 0.0022$ | $0.5041 \pm 0.0045$ |
| | | SGC | $0.9125 \pm 0.0025$ | $0.5056 \pm 0.0041$ |
| | | GPRGNN | $0.9512 \pm 0.0032$ | $0.5034 \pm 0.0051$ |
| | Facebook | GCN | $0.6775 \pm 0.0034$ | $0.5031 \pm 0.0046$ |
| | | GAT | $0.6443 \pm 0.0036$ | $0.5026 \pm 0.0043$ |
| | | SGC | $0.6280 \pm 0.0037$ | $0.5045 \pm 0.0047$ |
| | | GPRGNN | $0.5630 \pm 0.0029$ | $0.5039 \pm 0.0044$ |
| | LastFM | GCN | $0.8364 \pm 0.0031$ | $0.5044 \pm 0.0039$ |
| | | GAT | $0.8689 \pm 0.0034$ | $0.5037 \pm 0.0042$ |
| | | SGC | $0.8327 \pm 0.0032$ | $0.5018 \pm 0.0043$ |
| | | GPRGNN | $0.8472 \pm 0.0037$ | $0.5040 \pm 0.0048$ |
| | Chameleon | NLGCN | $0.6745 \pm 0.0060$ | $0.5132 \pm 0.0063$ |
| | | NLGAT | $0.5965 \pm 0.0069$ | $0.4849 \pm 0.0066$ |
| | | NLMLP | $0.4923 \pm 0.0077$ | $0.5357 \pm 0.0054$ |
| | | GPRGNN | $0.6541 \pm 0.0064$ | $0.5145 \pm 0.0053$ |

Table 19: Ablation Study of TSD's Defense performance against MIA. Continuation of Appendix Table 18.

| Method | Dataset | GNN Models | Classify Acc | Attack AUROC |
|---|---|---|---|---|
| TSD (Two Stage & Flattenning) | PubMed | GCN | $0.8381 \pm 0.0023$ | $0.4997 \pm 0.0048$ |
| | | GAT | $0.8400 \pm 0.0028$ | $0.4981 \pm 0.0061$ |
| | | SGC | $0.8080 \pm 0.0020$ | $0.5005 \pm 0.0057$ |
| | | GPRGNN | $0.8553 \pm 0.0014$ | $0.4987 \pm 0.0034$ |
| | Computers | GCN | $0.8845 \pm 0.0034$ | $0.4996 \pm 0.0042$ |
| | | GAT | $0.9122 \pm 0.0036$ | $0.5005 \pm 0.0041$ |
| | | SGC | $0.8364 \pm 0.0032$ | $0.4989 \pm 0.0043$ |
| | | GPRGNN | $0.8918 \pm 0.0029$ | $0.5013 \pm 0.0039$ |
| | Photo | GCN | $0.9301 \pm 0.0030$ | $0.4987 \pm 0.0037$ |
| | | GAT | $0.9432 \pm 0.0032$ | $0.4992 \pm 0.0046$ |
| | | SGC | $0.9089 \pm 0.0038$ | $0.5008 \pm 0.0040$ |
| | | GPRGNN | $0.9491 \pm 0.0027$ | $0.5006 \pm 0.0034$ |
| | Facebook | GCN | $0.6744 \pm 0.0034$ | $0.4981 \pm 0.0037$ |
| | | GAT | $0.6426 \pm 0.0029$ | $0.4979 \pm 0.0031$ |
| | | SGC | $0.6256 \pm 0.0037$ | $0.4990 \pm 0.0045$ |
| | | GPRGNN | $0.5609 \pm 0.0040$ | $0.4985 \pm 0.0039$ |
| | LastFM | GCN | $0.8340 \pm 0.0031$ | $0.4994 \pm 0.0041$ |
| | | GAT | $0.8672 \pm 0.0032$ | $0.4999 \pm 0.0042$ |
| | | SGC | $0.8325 \pm 0.0039$ | $0.4983 \pm 0.0045$ |
| | | GPRGNN | $0.8441 \pm 0.0027$ | $0.5005 \pm 0.0036$ |
| | Chameleon | NLGCN | $0.6657 \pm 0.0062$ | $0.4954 \pm 0.0065$ |
| | | NLGAT | $0.6585 \pm 0.0070$ | $0.4902 \pm 0.0063$ |
| | | NLMLP | $0.4824 \pm 0.0074$ | $0.5248 \pm 0.0051$ |
| | | GPRGNN | $0.6550 \pm 0.0058$ | $0.4956 \pm 0.0049$ |

