# OpenReview forum: "Improving Defense Mechanisms for Subgraph-Structure Membership Inference Attacks"
_ICLR.cc/2025/Conference — Submitted to ICLR 2025_

### Official Review · Reviewer_PGLU · 2024-10-29

**Soundness:** 3
**Presentation:** 3
**Contribution:** 3
**Rating:** 5
**Confidence:** 3

**Summary:**

The paper focuses on the defense strategies for graph neural networks against subgraph-structure membership inference attacks (SMIAs).Specically, the paper defend against the state-of-the-art SMIAs using flattening. The authors also strengthen the current attacks. Extensive results demonstrate that the proposed defense method can decrease the attack performance.

**Strengths:**

- Trendy topic
- Well-written paper

**Weaknesses:**

- Unclear motivation
- Performance is not that good
- Lack of adaptive attacks

**Questions:**

The paper proposed novel defenses and attacks on the domain of subgraph-structure membership inference attacks against graph neural networks. The authors first introduce the limitations of the previous state-of-the-art attacks. Then, the authors introduced the novel defense methods to defend against the attacks. Finally, the authors also improved the attacks. Experiences are conducted to demonstrate the effectiveness of the proposed defenses and attacks. The paper is well-written. However, I have several comments regarding motivation and performance.

- The authors proposed both defenses and attacks. The defenses can also mitigate the proposed new attacks. Therefore, I am wondering what is the motivation of the proposed attacks. The authors are supposed to explain why the proposed new attacks are important and what conclusions can be achieved by the new attacks.
- The defense method does not work that well. The attack success rate is still high enough to pose a threat. Therefore, the authors are suggested to discuss why the attacks can still achieve moderate performance.
- Instead of introducing new attacks, I recommend that the authors consider adaptive attacks against the proposed defenses. i.e., what can the adversary do if they know the defense methods?

---

> ### Author Response · Authors · 2024-11-24
>
> For Questions:
>
> 1. Our proposed new attack approach is as follows: the original SMIA attack is a non-end-to-end similarity-based attack, which performs well primarily on homophilic datasets and depends on the choice of similarity metrics. To evaluate our method comprehensively and on more diverse datasets and different model architectures, we also had to improve the performance of the original SMIA attacks on datasets the authors of the original paper did not consider. In a nutshell, we tested the original SMIA attack on our model, observed that it does not perform well (Please see the results in Table1,2 and Figure 3), and then proposed an improved attack – all in order to be fair and proper in our evaluations of the defense applied to more diverse settings than reported in the original SMIA work.
> At the same time, SMIA attacks is also an active research area. We hope that our proposed new attack method will inspire other researchers in the SMIA field, contributing to advancements in attack methodologies. We proposed the end-to-end learnable SMIA attack, which represents the first attempt to incorporate learnable permutation-invariant transformations into the field of SMIA. And our SMIA attack consistently outperforms existing baselines across various graph types, demonstrating superior adaptability and robustness. This advancement underscores the critical importance of an end-to-end learnable design in advancing the effectiveness of SMIA attacks, setting a new benchmark for future research in graph neural network security.
>
> 2. Thank you for your insightful observation. This issue largely arises from our assumption of an overly strong attack. Please refer to our response to your third question for in-depth clarifications. To demonstrate the defensive capabilities of TSD, in the experiments detailed in Section 6.1.1, we ensured that the shadow models employed by the attacker utilize the same model architecture, dataset, and dataset partitioning as the target model. It is important to note that the attacker is unaware that the dataset chosen for training the shadow model coincidentally matches that of the target model. This setup implies that the attacker possesses an exceptionally thorough understanding of the target model's training process, which is unrealistic in practical scenarios. In reality, an attacker would typically only be aware of the target model's functionality—for example, that it is used for disease prediction—and, at most, have knowledge of the general attributes of the training dataset (e.g., knowing that the target model uses disease-related data). Consequently, the experimental results pertain to nearly the strongest possible attacks, whereas the actual attack encountered in real-world applications would likely be less informed. Please also refer to the experiments in Section 6.1.2, where the dataset used to train the attacker's shadow models differs from that of the target model, resulting in a substantial decrease in Attack AUROC (For instance, when the target dataset is Facebook and the shadow dataset is CiteSeer, the AUROC of the End2end SMIA attack after applying our TSD defense method is only 0.5677). It is important to note that in the experimental setup of Section 6.1.2, the shadow model architecture remains identical to that of the target model, which may also be unlikely to see in practice. This indicates that the attack intensity in Section 6.1.2 remains relatively high. Furthermore, it is essential to recognize that Attack AUROC should not be minimized beyond random chance; an optimal defense mechanism should aim to reduce the attack success rate to approximately 50%, corresponding to random guessing.

---

> ### Author Response · Authors · 2024-11-24
>
> For Question:
>
> 3. Thank you very much for mentioning adaptive attacks (we never intended to examine this direction since we believe that conducting an adaptive attack against our method is quite challenging, but plan to add results in this direction). The first scenario of an adaptive attack is where the attacker knows that the target model employs the TSD defense method and is aware of the target model's architecture. Additionally, the attacker coincidentally selects the same dataset to train a shadow model. However, they are completely unaware of how the training and testing sets are partitioned within TSD (neither the proportion nor the partitioning method is known), making the thresholds used for Flattening and the test set in the two-stage training unknown. In this case, it is difficult for the attacker to adjust their attack strategy to better defend against the TSD method, even if they know that the defense mechanism is TSD. If the attacker attempts to better simulate the target model by training the shadow model using the same approach as TSD, any significant differences in the partitioning of training and testing sets between the shadow and target models would actually weaken the effectiveness of the attack. Under such circumstances, the best strategy for the attacker is to train the shadow model in a standard manner without applying TSD.
> The second scenario of an adaptive attack is where the attacker, in addition to the first case, also knows the target model's method for partitioning the dataset (though they still do not know the actual dataset used by the target model and have merely coincidentally chosen the same one). In this situation, if the attacker trains the shadow model in a standard manner, the results will be consistent with those in Tables 1 and 2 of Section 6.1.1, (since the experimental setup in Section 6.1.1 aligns with the second scenario of the adaptive attack). However, for the case that the attacker employs the same training strategy as TSD, we are currently attempting to conduct experiments under such scenarios and will present the results if you consider this setting to constitute an adaptive attack. If we cannot get the results by the end of rebuttal due to the time limit, we can put the results in revision.

---

### Official Review · Reviewer_8qXu · 2024-11-04

**Soundness:** 3
**Presentation:** 3
**Contribution:** 3
**Rating:** 6
**Confidence:** 5

**Summary:**

This paper discusses subgraph structure membership inference attacks (SMIA) on GNNs, which is an important yet unattended problem, and presents a novel defense mechanism. Specifically, for the attack method, the paper introduces an end-to-end SMIA attack model based on multiset functions, which seems to be a powerful attacker due to the order-invariant property. For the defense method, the paper designs a two-stage training strategy and a flattening technique to alter the posterior distributions and enhance defense capabilities.

**Strengths:**

- The paper is well written and technically sound.
- The attention on defenses on SMIA should be valued.
- The experiment are conducted thoroughly.

**Weaknesses:**

I have the following concerns about the settings of attacker and the effectiveness of the defense methods.

**Questions:**

*  For SMIA, the paper assumes that the attacker can access sufficient posterior distributions for an effective attack. However, it limits the range of application of this paper as that attackers might not be always available for obtaining such posterior information.

* In the defense, TSD uses soft labels to change the loss distribution, assuming that this label smoothing works well in all situations. However, for some tasks, using pseudolabels instead of actual labels might lower the model’s performance or cause unexpected biases.

* TSD assumes that it can remain effective across various types of GNNs and datasets. I am curious whether TSD can maintain the same level of performance on heterogeneous (not heterophilic) graphs is questionable.

---

> ### Author Response · Authors · 2024-11-24
>
> We thank the reviewer for the comment on the paper. However, we respectfully disagree with some points and would like to clarify as follows.
>
> For Questions:
>
> 1. Thank you for your question. In fact, we do not assume that the attacker requires a large number of posteriors for an effective attack. For details on the SMIA implementation process, please refer to Appendix B, Section "Standard SMIA Process." During the training of the attack model, the attacker uses publicly available datasets to train a shadow model, and this process does not require any posteriors generated from the target. When launching the attack, the attacker will only attempt to extract information on specific targets of interest. For example, if the attacker wants to determine whether a particular family has genetic/familial diseases, they might gain access to a medical data network and only input this family’s information into the network to obtain the relevant posterior, and use the attack model to extract private information from it (this is the most fundamental assumption in the research fields of MIA and SMIA. For specific details, please also refer to the original foundational articles [1]). In this case, the attacker only needs the posterior related to their target, not a large number of posteriors. In fact, our work follows the standard MIA and SMIA assumptions/procedures, which have been used by other researchers in the field as well.
>
> 2. Your observation is correct; using pseudolabeling does reduce the classification performance of the target model. We acknowledge this limitation and are actively exploring alternative approaches to mitigate its impact. Specifically, we are investigating strategies such as selectively sampling the test set based on confidence levels or randomly sampling nodes within the test set instead of utilizing the entire set. These methods offer a more effective trade-off compared to naively adding random noise, enabling us to achieve significantly better balances between maintaining classification accuracy and enhancing defense capabilities. We will continue to rigorously validate and enhance our research, and we appreciate your ongoing interest and attention to our work.
>
> 3. We thank the reviewer for the question. We only investigated it in the context of homophilic and heterophilic datasets. We believe that in the case of heterogeneous the TSD is still work but are in the process of running some additional experiments that we hope to report before the deadline. If we cannot report the results by the end of rebuttal due to the time limit, we can put the results in revision.
>
> [1] Membership Inference Attacks against Machine Learning Models

---

> > ### Comment · Reviewer_8qXu · 2024-11-24
> >
> > I have carefully reviewed the authors’ rebuttal and believe they have effectively addressed my concerns regarding Question 1. For Questions 2 and 3, I suggest including additional discussion to address potential concerns from the audience.

---

### Official Review · Reviewer_uYLi · 2024-11-06

**Soundness:** 3
**Presentation:** 3
**Contribution:** 2
**Rating:** 3
**Confidence:** 3

**Summary:**

This paper proposes a defense strategy against SMIA attacks. To defend sub-graph and label inference from GNNs, this paper devises a two-stage defense method (TSD) consisting of a flattening strategy and test set adjustment strategy.  The flatten strategy aims to introduce noises to the label distribution and the test set adjustment will reduce differences in outputs between the training set and the test set.

**Strengths:**

1. Sufficient empirical comparisons. Authors present lots of comparisons with other defense baselines to demonstrate advantages of TSD.
2. Clear writing and easy-to-follow.

**Weaknesses:**

1. Significant performance deterioration. Especially in Table3, after TSD defense, clean performance deteriorates from 80\% to 77\%. Performance deterioration problem is a significant problem in studies of defending against MIA attacks. I suggest authors also focus on tackling this problem.
2. Unclear motivation. The two-stage method is described clearly, but the motivation is unclear. For instance, Flatten strategy aims to introduce noises to the label distribution, but why could this help defend SMIA? In the second stage, does the test set adjustment aim to reduce differences in outputs between the training set and test set to defend against SMIA? If so, almost all defense motivation follows this motivation. Hence, I suggest more description or demonstration of motivations.

**Questions:**

Please refer to weaknesses.

---

> ### Author Response · Authors · 2024-11-24
>
> We thank the reviewer for the comment on the paper. However, we respectfully disagree with some points and would like to clarify as follows.
>
> For Weakness:
>
> 1、We respectfully disagree with the assertion. We have achieved better performance and tradeoff compared to other methods. For example, in the experimental scenarios presented in Table 1, our method achieves a 14.30% improvement in defense capability and a 10.05% enhancement in clear performance compared to existing methods. Similarly, in Table 2, our approach yields a 13.89% increase in defense capability and a 10.05% improvement in clear performance. These results demonstrate the significant advancements our method provides in both enhancing defensive measures against SMIA and maintaining robust classification accuracy. Besides, the drop of clear performance is inevitable. For reference, in one of the founding work "Membership Inference Attack on Graph Neural Networks", TABLE IV shows that when using their defense method, the average classification accuracy across datasets drops by 8.84% compared to the no-noise scenario. It is recognized in the field that such trade-offs between performance degradation and resistance to attacks are inevitable. Moreover, in comparison with no defense in Table 3, our approach demonstrates high effectiveness by achieving a 13% improvement in defense capability while only incurring a 3% reduction in classification accuracy. This trade-off underscores the superior performance of our method, highlighting its ability to enhance defense mechanisms significantly with minimal impact on classification performance. We will keep exploring better defense methods that can achieve better trade-offs, but this should not be the reason our contribution here is estimated.
>
> 2、Thank you for your question about the motivation. TSD is a two-stage process. In Section 4, M1 denotes the GNN model trained on the training set during the first stage (classical training), while M2 denotes the GNN model subsequently trained on the test set during the second stage, using the model M1 via test label predictions and initialization. More precisely, the parameters of M2 are initialized using M1 (i.e., M2 is a fine-tuned version of M1). Both stages of training employ Flattening; however, the Flattening in the second stage is applied more intensely, as we only apply Flattening when the training loss falls below a predefined threshold and in the second stage, the loss is easier to decrease below this threshold (because M2 is initialized by M1’s parameters). This indicates that Flattening is conducted for a longer duration and to a greater extent during the second stage. Hence, the two-stage process allows nonuniform flattening of the direct and indirect models. Aside from Flattening, the training of M1 follows a standard GNN process and effectively captures information about nodes and topology within the training set, enabling it to perform well for node classification tasks. However, for the same reason, M1 is highly vulnerable, with a high success rate for SMIA attacks. To utilize M1's strong node classification capabilities while preventing privacy leakage due to SMIA attacks, we refrain from directly using M1 and instead employ it indirectly. First, we flatten the training of M1, thereby modifying it parameters without significantly compromising accuracy while adding some limited protection. As evidenced by the results in Section 6.1.3, compared to Standard Training, Flattening approach results in an average decrease in accuracy <1%, and an average reduction of 4% in Attack AUROC. In the second stage, we train a new model in order to avoid using M1 directly. Instead, we use M1 to generate pseudo-labels for the test set and then train on the test set, further modifying M1 and thereby further increasing the defense. The Ablation Study comparison between Standard Training and Two-Stage (without Flattening) shows an average decrease of ~2% in node classification accuracy and an average defense improvement of ~9%. During the training of M2, we also apply Flattening, in a more aggressive manner. According to the Ablation Study, compared to Standard Training, Two-Stage & Flattening experiences a 3% drop in classification accuracy but improves the defense by 9%. Modifications to M1 can be performed in multiple rounds, using the same process. For example, the training and test sets can be redefined, allowing M2 to be fine-tuned with Flattening on a new training set to obtain M3, which is then used to generate pseudo-labels for the new test set, followed by training on the test set to obtain M4, etc. Such modifications trade-off node classification accuracy and defense capability, but have high complexity if more than two rounds are used. Overall, our method achieves better trade-off between complexity, node classification accuracy and defense capability.

---

### Official Review · Reviewer_oR7L · 2024-11-12

**Soundness:** 2
**Presentation:** 3
**Contribution:** 2
**Rating:** 3
**Confidence:** 4

**Summary:**

In this paper, the authors explore the robustness of graph neural networks (GNNs) against membership inference attack (MIA), specifically subgraph-structure membership inference attack (SMIA) where the foal is to infer whether a set of nodes form a particular structure e.g., a clique or multi-hop path in the training graph. The authors propose an SMIA attack which uses multiset function to improve the impact of the attack. To combat the attack, they propose a two-stage defense strategy. In the first stage, the model is trained on the training set, with some one-hot labels converted to soft labels to add ambiguity to selected samples. In the second stage, this model is further refined on the test set, where pseudo-labels are generated based on the model’s predictions after the first stage of training. Experimental results are included to support the claims.

**Strengths:**

1. The paper is well written.
2. Substantial experimental results are included in the paper.

**Weaknesses:**

1. The methodology section of the paper is lacking. The authors include comparison with other attacks and defenses in the methodology section which is not recommended. This section of the paper should emphasize on the proposed solution and the novelty. It is not clear as to why the new attack is more effective. Some explanations should be provided.

2. Novelty of the paper is lacking. Flattening is used by other defenses too [1].

3. For the proposed defense solution, the authors predict the pseudo-labels for the test nodes which is then used for training the model further. What happens when we have a dynamic graph where new nodes are added? Will the defense be still effective?

4. No adaptive attack is developed against the proposed defense solution to show its effectiveness.

[1] RelaxLoss: Defending Membership Inference Attacks without Losing Utility

**Questions:**

1. The problem formulation is not very clear. If I have the adjacency matrix of the graph available, why do I need to create a model to predict if the nodes form a particular structure or not? What setup is used for the experiments here - transductive or inductive?

2. Why small scale datasets are used for the SMIA attack? More statistical information about the dataset (structure you are trying to find) should be included to show the motivation of the work.

3. What kind of adaptive attack can be developed against the proposed solution?

---

> ### Author Response · Authors · 2024-11-24
>
> We thank the reviewer for the comment on the paper. However, we respectfully disagree with some points and would like to clarify as follows.
>
> For Weakness:
> 1. We believe that our work includes a detailed description of the methodology used. Section 4, "The TSD Method," provides a comprehensive explanation of our proposed TSD defense method against SMIA and MIA, highlighting the innovation of the TSD approach: two-stage training. This section also demonstrates the effectiveness of the TSD method by comparing the distributions of the training and test losses. Section 5 introduces a new attack method and explains why the newly proposed multiset approach enhances attack efficacy: the original SMIA employs 3 fixed similarity metrics, which not only forcibly segments the SMIA training process into two distinct parts but also lacks flexibility, preventing it from adapting to varying data distributions. Consequently, its ability to capture the structural information of the dataset is diminished, particularly in heterophilic datasets. But the multiset approach incorporates learnable permutation-invariant transformations, which allows for an end-to-end attack, improving its ability to capture more structural information (For example, in Tables 1 and 2, on the heterophilic dataset Chameleon, the Attack AUROC of the original SMIA is on average 22% lower in the 3-SMIA scenario and 24% lower in the 4-SMIA scenario compared to our proposed SMIA). Due to space limitation, we add all the necessary details about our method, but we will try to add more description on this part in our revision. And the methodology overview is also explained clearly via examples in Figure 1 and the caption.
> More intuition/motivation regarding TSD is provided below:
> TSD is a two-stage process. In Section 4, M1 denotes the GNN model trained on the training set during the first stage (classical training), while M2 denotes the GNN model subsequently trained on the test set during the second stage, using the model M1 via test label predictions and initialization. More precisely, the parameters of M2 are initialized using M1 (i.e., M2 is a fine-tuned version of M1). Both stages of training employ Flattening; however, the Flattening in the second stage is applied more intensely, as we only apply Flattening when the training loss falls below a predefined threshold and in the second stage, the loss is easier to decrease below this threshold (because M2 is initialized by M1’s parameters). This indicates that Flattening is conducted for a longer duration and to a greater extent during the second stage. Hence, the two-stage process allows nonuniform flattening of the direct and indirect models. Aside from Flattening, the training of M1 follows a standard GNN process and effectively captures information about nodes and topology within the training set, enabling it to perform well for node classification tasks. However, for the same reason, M1 is highly vulnerable, with a high success rate for SMIA attacks. To utilize M1's strong node classification capabilities while preventing privacy leakage due to SMIA attacks, we refrain from directly using M1 and instead employ it indirectly. First, we flatten the training of M1, thereby modifying it parameters without significantly compromising accuracy while adding some limited protection. As evidenced by the results in Section 6.1.3, compared to Standard Training, Flattening approach results in an average decrease in accuracy <1%, and an average reduction of 4% in Attack AUROC. In the second stage, we train a new model in order to avoid using M1 directly. Instead, we use M1 to generate pseudo-labels for the test set and then train on the test set, further modifying M1 and thereby further increasing the defense. The Ablation Study comparison between Standard Training and Two-Stage (without Flattening) shows an average decrease of ~2% in node classification accuracy and an average defense improvement of ~9%. During the training of M2, we also apply Flattening, in a more aggressive manner. According to the Ablation Study, compared to Standard Training, Two-Stage & Flattening experiences a 3% drop in classification accuracy but improves the defense by 9%. Modifications to M1 can be performed in multiple rounds, using the same process. For example, the training and test sets can be redefined, allowing M2 to be fine-tuned with Flattening on a new training set to obtain M3, which is then used to generate pseudo-labels for the new test set, followed by training on the test set to obtain M4, etc. Such modifications trade-off node classification accuracy and defense capability, but have high complexity if more than two rounds are used. Overall, our method achieves better trade-off between complexity, node classification accuracy and defense capability.

---

> ### Author Response · Authors · 2024-11-24
>
> For Weakness:
>
> 2. In this study, we make several significant contributions to the field of SMIA and their corresponding defenses. Firstly, we develop a multiset-based SMIA attack method that transforms the attack into an end-to-end learnable framework, thereby substantially enhancing its performance compared to existing approaches. This advancement enables more effective exploitation of structural information within graph neural networks. Secondly, we introduce a two-stage training procedure that is pivotal to our defense strategy. This procedure is the primary driver behind the notable improvements in defense performance, effectively mitigating the risk of privacy leakage through SMIA. Additionally, we incorporate Flattening as an auxiliary enhancement to our framework. Flattening serves as an extra boost, seamlessly integrating with our two-stage training process to further strengthen the defense without compromising the overall system compatibility. Please refer to Section 6.1.3, our ablation experimental results reveal that Flattening (One-Stage) leads to an average improvement of 4% in defense performance compared to Standard Training. Furthermore, when comparing Standard Training with Two-Stage (without Flattening), the two-stage approach achieves a more substantial average enhancement of 10% in defense effectiveness. These findings clearly demonstrate that the contribution of flattening to the defense capability is significantly smaller than that of two-stage training. What’s more, we did not claim Flattening as our main contribution and have already cited the Relaxloss paper. In conclusion, our contributions collectively advance the state-of-the-art in both attacking and defending against SMIA. The effectiveness of our proposed methods is thoroughly validated through comprehensive experiments, underscoring their practical applicability and superiority in various dataset scenarios.
>
> 3. Current research on SMIA and MIA defenses of GNNs does not involve dynamically changing graphs, although this problem is definitely of interest (thank you for bringing it up). In our paper, what we considered is the standard transductive graph learning problem. So there is no dynamics in graphs. This could be a topic of another interesting line but beyond the scope of our study. However, we believe that TSD would still be effective for defending against SMIA and MIA in dynamic graphs. This is because, whether the graph is dynamic or static, our TSD method is designed to modify the vulnerable model and change the loss distribution, thus providing robust defense against both SMIA and MIA.
>
> 4. Thanks for mention adaptive attack. We have thoroughly considered adaptive attacks. A detailed information of our thought about adaptive attacks can be found in our response to your third question.
>
> [1] Membership Inference Attacks against Machine Learning Models
> [2] Subgraph Structure Membership Inference Attacks against Graph Neural Networks

---

> ### Author Response · Authors · 2024-11-24
>
> For Question:
> 1. In the "Problem Formulation", the graph dataset G is defined as the dataset used by the GNN user (the victim) for training. Naturally, the GNN user has full knowledge of this dataset, including all node features, adjacency matrices, etc. On the other hand, the GNN is a black box for attackers, which means when the attacker targets the GNN trained by the user, they have no knowledge of the user’s training dataset G, but only have access to use the GNN (this is one of the most common assumption in the research fields of MIA and SMIA – see foundational articles [1] and Appendices B and C). Therefore, attackers need to rely on a shadow dataset Gs (also defined in the "Problem Formulation") to train a shadow model, aiming to infer possible membership and structural information about nodes in G. In summary, the target model and attack model may be trained on different graphs because the attacker do not know the knowledge of GNN user’s dataset G. Specifically, while the GNN user has access to the entire graph topology, this does not render it meaningless for the attacker to design an attack model aimed at inferring the graph topology, as the attacker does not have knowledge of it. Please see [2] for more details, as our problem formulation is identical to that presented in [2], which is an already published paper. In this paper, we conduct experiments using the transductive setting.
>
> 2. In our SMIA experiments, we utilized datasets that are commonly tested in MIA/SMIA-related research (see [2]). The structural statistics of these datasets are as follows:
>
> CiteSeer: 27,174 2-hop paths,  1,547 3-cliques,  185,706 3-hop paths,  514 4-cliques;
>
> Facebook: 17,321,238 2-hop paths,  1,723,172 3-cliques,  33,284,821 3-hop paths,  16,478,329 4-cliques;
>
> LastFM: 679,080 2-hop paths,  40,433 3-cliques,  1,201,750 3-hop paths,. 65,442 4-cliques;
>
> Chameleon: 523,492 2-hop paths,  29,342 3-cliques,  1,123,528 3-hop paths,  57,381 4-cliques;
>
> These structural statistics provide a comprehensive overview of the complexity and diversity of the datasets used in our SMIA experiments, facilitating a robust evaluation of the proposed defense methods. In comparison to [2], we have additionally incorporated a heterophilic dataset Chameleon to demonstrate the effectiveness of our defense method, TSD, and our attack method, End2end SMIA. Regarding larger datasets, the substantial increase in the number of substructures combined with limited computational resources made the experiments exceedingly time-consuming. Consequently, we did not utilize large-scale datasets such as ogbn-arxiv for conducting SMIA defense experiments. However, for MIA defense, we did employ the ogbn-arxiv dataset (see Appendix Tables 13 and 14). These results underscore the robustness of our proposed method across diverse dataset structures while avoiding practical limitations encountered in connection with large datasets.
>
> [1] Membership Inference Attacks against Machine Learning Models
> [2] Subgraph Structure Membership Inference Attacks against Graph Neural Networks

---

> ### Author Response · Authors · 2024-11-24
>
> For Question:
>
> 3. Thank you very much for mentioning adaptive attacks (we never intended to examine this direction since we believe that conducting an adaptive attack against our method is quite challenging, but plan to add results in this direction). The first scenario of an adaptive attack is where the attacker knows that the target model employs the TSD defense method and is aware of the target model's architecture. Additionally, the attacker coincidentally selects the same dataset to train a shadow model. However, they are completely unaware of how the training and testing sets are partitioned within TSD (neither the proportion nor the partitioning method is known), making the thresholds used for Flattening and the test set in the two-stage training unknown. In this case, it is difficult for the attacker to adjust their attack strategy to better defend against the TSD method, even if they know that the defense mechanism is TSD. If the attacker attempts to better simulate the target model by training the shadow model using the same approach as TSD, any significant differences in the partitioning of training and testing sets between the shadow and target models would actually weaken the effectiveness of the attack. Under such circumstances, the best strategy for the attacker is to train the shadow model in a standard manner without applying TSD.
> The second scenario of an adaptive attack is where the attacker, in addition to the first case, also knows the target model's method for partitioning the dataset (though they still do not know the actual dataset used by the target model and have merely coincidentally chosen the same one). In this situation, if the attacker trains the shadow model in a standard manner, the results will be consistent with those in Tables 1 and 2 of Section 6.1.1, (since the experimental setup in Section 6.1.1 aligns with the second scenario of the adaptive attack). However, for the case that the attacker employs the same training strategy as TSD, we are currently attempting to conduct experiments under such scenarios and will present the results if you consider this setting to constitute an adaptive attack. If we cannot get the results by the end of rebuttal due to the time limit, we can put the results in revision.

---

### Comment · Area_Chair_QL1W · 2024-11-28

I would like to encourage the reviewers to engage with the author's replies if they have not already done so. At the very least, please
acknowledge that you have read the rebuttal.

---

### Meta-Review · Area_Chair_QL1W · 2024-12-19

**Metareview:**

This paper studies subgraph-structure membership inference attacks (SMIA). it is not clear to me how relevant are SMIA in real-world settings, and I find some of the scenarios used for motivation unconvincing, nonetheless, this problem is interesting from a technical point of view. The paper proposes a two-stage defence where the first model is trained on the training set with flattening (introduced in prior work), then used to generate pseudo-labels for a second model trained on test set. The authors also propose a SMIA attack model using multisets (Deep Sets). Some reviewers found the motivation and methodology description unclear. In the initial submission the authors did not introduce any adaptive attacks against the proposed defence. In the rebuttal the authors provided additional experiments trying to address the issue with adaptive attacks, however, these attacks are not very convincing. For example the claim "the best strategy for the attacker is to train the shadow model in a standard manner without applying TSD." in the first scenario needs to be better substantiated.

In future iterations I suggest the authors to better motivate the two-stage approach and consider additional improvements to reduce the accuracy drop. More importantly, I suggest the authors to spend more time on developing stronger adaptive attacks given that the defences is heuristic. These attack should be the main focus (part of the main paper rather than the appendix). Experiments on larger-scale datasets to validate scalability would also straighten the paper.

**Additional Comments On Reviewer Discussion:**

The reviewers raised points during discussion regarding the methodology clarity and motivation, adaptive attacks, and performance trade-offs.  The authors clarified that the two-stage process allow to better balance privacy and utility. The ablation studies shows some evidence that both components are needed. A drop in accuracy is expected, and the authors claims that their method achieves better trade-offs than prior work. Prompted by Reviewer 8qXu the authors included additional experiments on DBLP and IMDB to validate the performance of the method on heterogeneous graphs.

---

### Decision · Program_Chairs · 2025-01-22

Reject